

# Origin of elemental carbon in snow from Western Siberia and northwestern European Russia during winter–spring 2014, 2015 and 2016

**Nikolaos Evangeliou**[1],*, **Vladimir P. Shevchenko**[2], **Karl Espen Yttri**[1], **Sabine Eckhardt**[1], **Espen Sollum**[1], **Oleg S. Pokrovsky**[3,4], **Vasily O. Kobelev**[5], **Vladimir B. Korobov**[6], **Andrey A. Lobanov**[5], **Dina P. Starodymova**[2], **Sergey N. Vorobiev**[7], **Rona L. Thompson**[1], **Andreas Stohl**[1]

[1] NILU - Norwegian Institute for Air Research, Department of Atmospheric and Climate Research (ATMOS), Kjeller, Norway.

[2] Shirshov Institute of Oceanology, Russian Academy of Sciences, Nakhimovsky prospect 36, 117997 Moscow, Russia.

[3] Geosciences Environment Toulouse, UMR 5563 CNRS, University of Toulouse, 14 Avenue Edouard Belin, 31400, Toulouse, France.

[4] N. Laverov Federal Center for Integrated Arctic Research, Russian Academy of Science, Sadovaya street, 3, 163000, Arkhangelsk, Russia.

[5] Arctic Research Center of the Yamalo-Nenets autonomous district, Vos'moy proezd, NZIA building, 629730, Nadym, Yamalo-Nenets autonomous district, Russia.

[6] North-Western Brunch of Shirshov Institute of Oceanology, Russian Academy of Sciences, Naberezhnaya Severnoy Dviny 112/3, 163061, Arkhangelsk, Russia.

[7] BIO-GEO-CLIM Laboratory, Tomsk State University, 36 Prospect Lenina, 634050, Tomsk, Russia.

*Correspondence to: N. Evangeliou, NILU - Norwegian Institute for Air Research, Department of Atmospheric and Climate Research (ATMOS), Kjeller, Norway (Nikolaos.Evangeliou@nilu.no)



## Abstract

Short–lived climate forcers have been proven important both for the climate and human health. In particular, black carbon (BC) is an important climate forcer both as an aerosol and when deposited on snow and ice surface, because of its strong light absorption. This paper presents measurements of elemental carbon (EC; a measurement-based definition of BC) in snow collected from Western Siberia and northwestern European Russia during 2014, 2015 and 2016. The Russian Arctic is of great interest to the scientific community due to the large uncertainty of emission sources there. We have determined the major contributing sources of BC in snow in Western Siberia and northwestern European Russia using a Lagrangian atmospheric transport model. For the first time, we use a recently developed feature that calculates deposition in backwards (so-called retroplume) simulations allowing estimation of the specific locations of sources that contribute to the deposited mass.

EC was found in higher levels compared to previously reported concentrations and it was highly variable depending on the sampling location. Modelled BC was in good agreement ($R = 0.53 - 0.83$) with measured EC. However, a systematic region–specific model underestimation was found. For EC sampled in northwestern European Russia the underestimation by the model was smaller (> -100%). In this region, the major sources were transportation activities and domestic combustion in Finland. When sampling shifted to Western Siberia, the model underestimation was more significant (< -100%). There, the sources included emissions from gas flaring as a major contributor to snow BC. The accuracy of the model calculations was also evaluated using two independent datasets of BC measurements in snow covering the entire Arctic. The model reproduced snow BC concentrations quite accurately, although small discrepancies occurred mainly for samples collected in springtime. Nevertheless, EC concentrations in snow presented here are about 20% lower than previously reported ones in Western Siberia and northwestern European Russia.



## 1    Introduction

Black carbon (BC) is the most strongly light-absorbing component of the atmospheric aerosol and is formed by the incomplete combustion of fossil fuels, biofuels, and biomass (Bond et al., 2013). It is emitted directly into the atmosphere in the form of fine particles. BC is a major component of "soot", a complex light-absorbing mixture that also contains organic carbon (OC) (Bond et al., 2004). Combustion sources emitting BC include open biomass burning (forest, savanna, agricultural burning), residential biofuel combustion, diesel engines for transportation or industrial use, industrial processes and power generation, or residential coal combustion (Liu et al., 2011; Wang et al., 2011).

The main reasons why BC is important on a global perspective are its impacts on human health and on climate. As a component of the fine particulate matter (PM2.5), it is associated with negative health impacts, including premature mortality (Lelieveld et al., 2015; Turner et al., 2005). It also absorbs solar radiation, has a significant impact on cloud formation and, when deposited on ice and snow, it absorbs radiation there and accelerates melting (Hansen and Nazarenko, 2004). BC has a lifetime that can be as long as 9–16 days (Bond et al., 2013). After its emission, BC can travel over long distances (Forster et al., 2001; Stohl et al., 2006) and reach remote areas such as the Arctic. Arctic land areas are covered by snow in winter and spring, while the Arctic Ocean is partly covered by ice. Sea ice has a much higher albedo (≈0.5–0.7) compared to the surrounding ocean (≈0.06), thus BC deposited on sea ice reduces the heat uptake of the ocean. Snow has an even higher albedo than sea ice and can reflect as much as 90% of the incoming solar radiation (Brandt et al., 2005; Singh and Haritashya, 2011).

Hegg et al. (2009) reported that snow in the Arctic often contains BC at concentrations between 1 and 30 ppb, which can cause a snow albedo reduction of 1–3% in fresh snow and another 3–9% as snow ages and BC becomes more concentrated near the surface (Clarke and Noone, 1985). This solar radiation reflecting capacity of snow insulates the sea ice, maintains cold temperatures and delays ice melt in summertime. After the snow begins to melt and because shallow melt ponds have an albedo of approximately 0.2 to 0.4, the surface albedo drops to about 0.75 or even lower (0.15) as melt ponds grow and deepen (Singh and Haritashya, 2011). These changes have been found to be important for the global energy balance (Flanner et al., 2007; Hansen and Nazarenko, 2004) and, if enhanced by BC, contribute to climate warming (Warren and Wiscombe, 1980).



Although BC in Arctic snow and ice has been found to be important for the Earth's climate (Flanner et al., 2007; Sand et al., 2015), its large-scale temporal and spatial distributions and exact origin are still poorly quantified (AMAP, 2015). Efforts to determine the concentrations of BC in snow across the Arctic were made by Clarke and Noone (1985), Doherty et al. (2010, 2013), (Forsström et al., 2013; Ingvander et al., 2013; McConnell et al., 2007). This paper presents measurements of Elemental Carbon (EC) concentrations in snow samples collected in spring 2014, 2015 and 2016 in the Kindo Peninsula (White Sea, Karelia), around Arkhangelsk in northwestern European Russia, and in Western Siberia. In the latter area, gas flaring emissions are very important. Flaring emissions are highly uncertain because both activity data and emission factors are largely lacking. According to the Global Gas Flaring Reduction Partnership (GGFR) (http://www.worldbank.org/en/programs/gasflaringreduction), nearly 50 billion m$^3$ of gas are flared in Russia annually. The Russian flaring emissions in the Nenets/Komi regions and in Khanty-Mansiysk are the major sources in the area. It has been reported that gas flaring in Russia contributes about 42% to the annual average BC surface concentrations in the Arctic (Stohl et al., 2013).

The use of the terms EC and BC has been the topic of several scientific papers (Andreae and Gelencsér, 2006; Bond et al., 2013; Petzold et al., 2013). Petzold et al. (2013) defined BC as a substance with 5 properties (see Table 1 in Petzold et al., 2013), for which no single measurement instrument exists that is sensitive to all of them at the same time. Consequently, BC cannot uniquely be measured, although some of its properties can, such as the absorption coefficient $\sigma_{ap}$ and the elemental carbon (EC) concentration, both commonly measured in atmospheric monitoring networks across the world. Hence, the term BC should be used in a qualitative manner. In the present study, EC measurement data from three campaigns are compared to simulation results from the Lagrangian particle dispersion model (LPDM) FLEXPART. The model is used here for the first time to quantify the sources contributing to BC in snow in Russia adopting a special feature that was developed recently.

## 2 Methodology

### 2.1 Collection and storage of snow samples

Snow samples were collected along a north–south transect between Tomsk and the Yamal coast in February–March 2014, while in March 2015 sample collection took place in the Kindo Peninsula and near the port of Arkhangelsk near the White Sea (Figure 1). Finally,



in February–May 2016 samples were collected in the Kindo Peninsula, in Arkhangelsk and
between Tomsk and Yamal. These areas have been reported to receive pollution both from
urban pollution and gas flaring sources (Stohl et al., 2013). For example, the gas flaring
sources located in Yamal and Khanty-Mansiysk (Russia) are in the main pathway along
which sub-Arctic air masses travel to the Arctic (Stohl et al., 2006). All sampling points were
located more than 500 m away from roads to minimize the influence from local traffic
emissions. Information about the samples such as the location of sampling, the amount of
snow collected and the depth to which snow was sampled is reported in Table S1 and the
sample locations are plotted in Figure 1.
Sampling was performed using a metal-free technique, using pre-cleaned plastic shovels
and single–use vinyl gloves. Samples were stored in polyethylene bags, which had been
thoroughly washed with 1 M HCl and with abundant Milli–Q water in the laboratory prior to
their use. After returning the samples to the laboratory, the snow was allowed to melt at
ambient temperature (18–20°C), and immediately filtered through quartz fibre filters (47 mm
diameter 2500QAT-UP, Pall, for samples collected in 2014 and 47 mm diameter QM-A
Whatman for samples collected in 2015 and 2016). The filters were dried at 60–70°C,
wrapped in aluminum foil and stored in a refrigerator.

## 137 2.2 Elemental Carbon measurements by Thermal–Optical Analysis (TOA)

The filters' content of elemental carbon (EC) was measured at NILU's laboratories by
thermal-optical analysis (TOA), using the Sunset laboratory OC/EC instrument operated
according to the EUSAAR-2 protocol (Cavalli et al., 2010). A 1.5 cm$^2$ punch was cut from the
filtered snow samples for the analysis. Transmission was used for organic carbon (OC)
charring correction. The OC/EC instrument's performance is regularly intercompared as part
of the joint European Monitoring and Evaluation Programme (EMEP)/Aerosols, Clouds, and
Trace gases Research InfraStructure Network (ACTRIS) quality assurance and quality control
effort (Cavalli et al., 2015).

## 146 2.3 Measurements of carbonate ($CO_3^{2-}$)–carbon by Thermal–Optical Analysis
## 147 (TOA) following thermal-oxidative pre-treatment

The content of carbonate ($CO_3^{2-}$)–carbon on the filters was measured by TOA,
following thermal-oxidative pretreatment based on the approach described by Jankowski et al.
(2008). In brief, a punch of 1.5 cm$^2$ from each filter was heated at 450 °C for 2 hours in



ambient air to remove OC and EC, but not $CO_3^{2-}$–carbon. The filter punch was subjected to
TOA immediately (30 sec) after thermal-oxidative pre-treatment. The split time (between OC
and EC) obtained for each filter punch used to determine the filter samples' content of EC
(Chapter 2.2) was also used to apportion $CO_3^{2-}$–carbon to OC and/or EC. The influence of
$CO_3^{2-}$–carbon evolving as EC, was accounted for by the following equation:

$$EC_{CO_3^{2-}}^{corr} = EC - EC_{CO_3^{2-}}$$

where $EC_{CO_3^{2-}}^{corr}$ is elemental carbon corrected for $CO_3^{2-}$–carbon that evolves as EC during
TOA, EC is elemental carbon and $EC_{CO_3^{2-}}$ is $CO_3^{2-}$–carbon that evolves as EC during TOA.
EC values were 5-22% lower applying this correction (see Supplementary Information).
**2.4   Emissions and modelling of black carbon**
The concentrations of BC in snow were simulated with the LPDM FLEXPART version
10 (Stohl et al., 1998, 2005). The model was driven with 3–hourly (for the years 2014 and
2015) and hourly (for the year 2016) operational meteorological wind fields retrieved from
the European Centre for Medium-Range Weather Forecasts (ECMWF). The ECMWF data
have 137 vertical levels. The data used had a horizontal resolution of 1°×1° for the 2014 and
2015 simulations and 0.5°×0.5° for the 2016 ones.
The simulations were conducted in backwards time ("retroplume") mode, using a new
feature of FLEXPART to reconstruct wet and dry deposition with backward simulations
(Eckhardt et al., 2017). This new feature is an extension of the traditional possibility to
simulate atmospheric concentrations backward in time (Seibert and Frank, 2004; Stohl et al.,
2003). It is computationally efficient because it requires only two single tracer transport
simulations (one for wet deposition, one for dry deposition) for each measurement sample. To
reconstruct wet deposition amounts of BC, computational particles were released at altitudes
of 0 to 20 km at the locations where snow samples were taken, whereas to reconstruct dry
deposition, particles were released between the surface and 30 m at these locations. All
released particles represent a unity deposition amount, which was converted immediately (i.e.,
upon release of a particle) to atmospheric concentrations using the deposition intensity as
characterized either by dry deposition velocity or scavenging rate (for further details, see
Eckhardt et al., 2017). The concentrations were subsequently treated as in normal
"concentration mode" backward tracking (Seibert and Frank, 2004) to establish source-
receptor relationships between the emissions and deposition amounts. The termination time of



the particle release was the time at which the snow sample was collected, whereas the
beginning time was set as the time when the ECMWF precipitation at the sampling site,
accumulated backward in time, was equal to the water equivalent of the snow sample, up to
the specified sampling depth.
The model output consists of a spatially gridded sensitivity of the BC deposition at the
sampling location (receptor) to the BC emissions, equivalent to the backwards time mode
output for concentrations (Seibert and Frank, 2004; Stohl et al., 2003). BC deposition at the
snow sampling point can be computed (in mass per area units) by multiplying the emission
sensitivity in the lowest model layer (the footprint emission sensitivity) with gridded
emissions from a BC emission inventory and integrating over the grid. The deposited BC can
be easily converted to BC snow concentration by taking into account the water equivalent
depth of the snow from ECMWF (in mm). In the present study, the ECLIPSE (Evaluating the
CLimate and Air Quality ImPacts of ShortlivEd Pollutants) version 5 emission inventory
(Klimont     et     al.,     2016;     Stohl     et     al.,     2015)     was     used
(http://www.iiasa.ac.at/web/home/research/researchPrograms/air/Global_emissions.html).
The total emissions of BC from ECLIPSE in the areas of study are shown in Figure 1 (left
panel).
BC was assumed to have a density of 2 g m$^{-3}$ in our simulations and a logarithmic size
distribution with an aerodynamic mean diameter of 0.25 μm and a logarithmic standard
deviation of 0.3. Each computational particle released in FLEXPART represents an aerosol
population with a lognormal size distribution (see Stohl et al., 2005). Assumed aerodynamic
mean diameter and logarithmic standard deviation are used by FLEXPART's dry deposition
scheme, which is based on the resistance analogy (Slinn 1982), and they are consistent with
those used in other transport models (see Evangeliou et al., 2016; Shiraiwa et al., 2008).
Below-cloud scavenging was determined based on the precipitation rate taken from ECMWF.
The in-cloud scavenging was based on cloud liquid water and ice content, precipitation rate
and cloud depth from ECMWF (Grythe et al. 2016). The FLEXPART user manual (available
from  http://www.flexpart.eu)  provides  more  information.  All  modelling  results  for  this
sampling     campaign     can     be     viewed     interactively     at     the     URL
http://niflheim.nilu.no/NikolaosPY/SnowBC_141516.py.
**3   Results**
**3.1   Elemental Carbon concentrations measured in snow**



The spatial distribution of EC measured in snow samples from northwestern European
Russia and Western Siberia is shown in Figure 1 for each of the campaigns (2014, 2015 and
2016). There was large spatial variability in the distribution of EC in snow in 2014 ranging
from 3 to 219 ng g$^{-1}$, with a median (±standard deviation) of 23±50 ng g$^{-1}$. The highest EC
concentrations in 2014 were observed in Western Siberia near Tomsk (147 to 219 ng g$^{-1}$).
FLEXPART emission sensitivities for these samples showed that the air was coming from the
north and the east (see in http://niflheim.nilu.no/NikolaosPY/SnowBC_141516.py). This
explains the high concentrations of EC, as most of the anthropogenic BC sources are located
in these regions. In the rest of the snow samples, EC concentrations between 4 and 170 ng g$^{-1}$
were observed. High concentrations were observed near the Ob River coinciding with air
masses arriving mainly from Europe. During the 2015 field campaign, EC concentrations
were the highest near Arkhangelsk (175 ng g$^{-1}$), for which FLEXPART showed that the air
was coming from nearby areas (http://niflheim.nilu.no/NikolaosPY/SnowBC_141516.py).
Therefore, it is likely that the samples were affected by direct emissions from the city or the
port of Arkhangelsk. During the same campaign, snow samples collected in the Kindo
peninsula (at the White Sea coast) showed high variability in EC concentrations (range: 46 –
152 ng g$^{-1}$, median=70±37 ng g$^{-1}$). According to FLEXPART emission sensitivities, air
masses were transported to Kindo peninsula from central and southern Europe driven by an
anticyclone over Scandinavia (http://niflheim.nilu.no/NikolaosPY/SnowBC_141516.py).
Finally, in 2016, snow samples were collected outside Arkhangelsk, at the Kindo peninsula,
as well as close to the Yamal Peninsula in Western Siberia (range: 7–161 ng g$^{-1}$, median:
40±39 ng g$^{-1}$). In Arkhangelsk, EC concentrations in snow varied widely in 2016 ranging
from 31 to 161 ng g$^{-1}$ with a median concentration in this region of 61±45 ng g$^{-1}$. This is far
below the 175 ng g$^{-1}$ observed in 2015, although there was only one sample collected in that
year. In the Kindo Peninsula, EC was relatively constant in 2016 ranging between 25 and 35
ng g$^{-1}$ (median = 28±4 ng g$^{-1}$), which is more than 60% lower compared with the 2015 values
(median = 70±37 ng g$^{-1}$). Finally, between Tomsk and Yamal EC was highly variable (7 – 119
ng g$^{-1}$) due to the different EC sources affecting snow (median = 50±38 ng g$^{-1}$). For instance,
it is expected that gas flaring affects snow close to Yamal, while snow collected in the south
(Tomsk) is likely influenced by sources in Europe or local urban emissions. Nevertheless, the
highest concentrations (>100 ng g$^{-1}$) were observed north of 68°N, in the Yamal Peninsula.
We compared the measured snow concentrations with those calculated by FLEXPART.
For this, the emission sensitivities were multiplied with the total emission fluxes from





ECLIPSE (section 2.4). A scatter plot is presented in Figure 1 (middle panel). The results
show good quantitative agreement and a good correlation between modelled BC and
measured EC concentrations for the 2015 and 2016 campaigns ($R_{2015} = 0.83$ and $R_{2016} = $
0.68, $p-value < 0.05$), but weaker correlation for 2014 ($R_{2014} = 0.53$, $p-value < $
0.05). For further validation, the fractional bias (FB) of each individual sample was calculated
together with the mean fractional bias (MFB) for observed EC and modelled BC for the 2014,
2015 and 2016 sampling campaigns as follows:

$$FB = \frac{C_m - C_o}{(C_m + C_o)/2} \times 100\% \ and \ MFB = \frac{1}{N} \sum_{i=1}^{N} \frac{C_m - C_o}{(C_m + C_o)/2} \times 100\%$$

where $C_m$ and $C_o$ are the modelled BC and measured EC concentrations and $N$ is the total
number of observations for each year. The FB for individual samples is shown in Figure S 1.
FB is a useful model performance indicator because it is symmetric and gives equal weight to
underestimations and overestimations (it takes values between -200% and 200%). It is used
here to show the locations where modelled BC concentrations in snow over- or underestimate
observations (see Figure S 1). The MFB of the model for the 2014 snow measurements was -
42%. In total, the model underestimated concentrations for 17 out of 23 samples with FB
values ranging from -168% to -30%, whereas for the rest (six samples) FB values ranged
between 20% and 148% (median: -56%±81%) (Figure S 1). In 2015, the MFB of the model
was -48% (median: -56%±32%), where 11 out of 12 values were underestimated by the model
showing FB values that ranged between -101% and -7% (one FB value was found to be 12%).
Finally, FB values of the simulated concentrations of BC in snow showed another
underestimation for 2016 (median: -13%±73%) varying from -198% to -0.3% for 12 out of
19, whereas for seven samples the model predicted higher concentrations compared with
observations (10% to 75%) (Figure S 1). Furthermore, the root mean square error (RMSE)
was computed, which is a frequently used measure of the differences between values
predicted by a model and the values actually observed. RMSE values were estimated to be
quite high, between 37 and 49 ng g$^{-1}$, due to the large variation of the observed EC
concentrations.
The levels of EC in snow presented here are relatively high compared to previously
reported concentrations in the Arctic. Excluding Aamaas et al. (2011) who reported EC
concentration in snow close to the Svalbard airport greater than 1000 ng g$^{-1}$, Ruppel et al.
(2014) found that EC concentrations have been increasing up to 103 ng g$^{-1}$ since 1970 in




Svalbard. McConnell et al. (2007) reported that the BC concentrations measured at the D4
ice-core site in Greenland were 10 ng g$^{-1}$, at maximum, which most likely originated from
biomass burning in the conifer–rich boreal forest of the Eastern and Northern United States
and Canada. Forsström et al. (2013) reported concentrations as high as 88 ng g$^{-1}$ in
Scandinavia, and lower ones at higher latitudes (11–14 ng g$^{-1}$ in Svalbard, 7–42 ng g$^{-1}$ in the
Fram Strait, and 9 ng g$^{-1}$ in Barrow). Svensson et al. (2013) collected snow samples from
Tyresta National Park and Pallas-Yllästunturi National Park in Sweden. Tyresta is a relatively
polluted site located circa 25 km from the city centre of Stockholm (population of about 2
million people). Yllästunturi National Park is located in Arctic Finland and a clean site with
no major city influencing the local and regional air. The concentration of EC in Pallas-
Yllästunturi was between 0 and 140 ng g$^{-1}$, while in Tyresta the BC concentrations were one
order of magnitude higher (53–810 ng g$^{-1}$). Furthermore, Doherty et al. (2010) in the most
complete dataset for the Arctic snow and ice BC reported highly variable concentrations (up
to 800 ng g$^{-1}$) for five consecutive years (2005–2009). Finally, in the most recent dataset for
snow BC (Macdonald et al., 2017), concentrations ranging from 0.3 to 15 ng g$^{-1}$ were reported
for samples collected near the Alert observatory (see section 4.1).
**3.2   Sources and origin of BC**
We further analysed the model output in order to calculate relevant contributions from
various BC source types to BC concentrations in snow (for method description, see section
2.4). ECLIPSE emissions account for waste burning (WST), industrial combustion and
processing (IND), surface transportation (TRA), power plants, energy conversion, and
extraction (ENE), residential and commercial combustion (DOM), gas flaring (FLR), whereas
biomass burning (BB) emissions were adopted from the Global Fire Emissions Database,
Version 4 (GFEDv4.1) (Giglio et al., 2013). The results are depicted in Figure 2 for the
sampling campaigns of 2014, 2015 and 2016 in Western Siberia and North-Western European
Russia, sorted from the northernmost to the southernmost sampling location.
In 2014, TRA contributed about 18%, on average, to the simulated BC in snow, DOM
28%, FLR 44%, whereas ENE and IND were less significant. Maxima of TRA, DOM, and
FLR contributions were observed at a latitude of about 65°N, where measured EC and
modelled BC were similar. An example of the contribution from the aforementioned
dominant sources to snow BC concentrations for the highest measured EC concentration in
snow is shown in Figure 3. The transport sector includes emissions from all land-based





transport of goods, animals and persons. It is more significant in southern Russia and close to
the borders with Kazakhstan and Mongolia, where a large number of major Russian cities are
located (e.g., Moscow, Kazan, Voronezh, Saratov, Samara, Ufa, Perm, Yekaterinburg,
Tyumen, Omsk, Tomsk, Novosibirsk, Krasnoyarsk, etc…) and connected with each other by
federal highways. Residential and commercial combustion includes emissions from
combustion in households and public and commercial buildings. Therefore, it is important for
areas with large population centres (Figure 3). FLR emissions were found to contribute the
most in this example with a total concentration from this sector of 19.7 ng $g^{-1}$ (relative to 12.6
and 16.5 ng g-1 in TRA and DOM, respectively) (Figure 3).
In the Kindo Peninsula and in Arkhangelsk, where snow sampling took place in 2015,
the main contributions to snow BC were from DOM (47%), TRA (30%), BB (7%), and FLR
(6%) (Figure 2). Similar to EC measurements in snow, simulated BC was also higher than in
2014, as the sampling sites were located closer to strong sources in Europe (Kindo) and close
to a populated area (Arkhangelsk) with a strong regional impact. The highest concentration of
EC was observed in the Kindo Peninsula (33.13°E – 66.53°N). Figure 4 shows the spatial
distribution of emissions that contributed to simulated snow BC at the sampling point where
the highest BC concentration was observed. In this case, TRA and DOM emissions from
Europe mostly affected snow in the Kindo Peninsula, whereas FLR emissions were very low
due to the large distance from the sampling point. Emissions from an unusual late winter/early
spring episode of BB in the borders of Belarus, Ukraine and Russia also affected snow
concentrations in northwestern European Russia (Figure 4). The importance of episodic BB
releases in Russia and the miscalculation of satellite retrieved BB emissions and their impact
in Arctic concentrations in early spring has been explained by Evangeliou et al. (2016) and
Hao et al. (2016). BB emissions contributed about 19.4 ng $g^{-1}$ to the snow concentration at the
receptor point, mostly originating from Eastern Europe (Figure 4). TRA and DOM emissions
were the dominant sources for this sampling point, contributing 33.6 and 47.2 ng $g^{-1}$,
respectively (Figure 4).
Finally, in 2016, when samples were collected at the Kindo Peninsula, in Arkhangelsk
and in Yamal, DOM contributed 31%, FLR about 29% and TRA 27%, on average (Figure 2,
bottom). Similar to the measured EC concentrations in snow, simulated concentrations of BC
in 2016 were lower than those in 2015, on average. The highest measured EC concentration
was observed in the Khanty-Mansiysk region (72.94°E – 65.36°N), which mirrors the
simulated BC concentration at the same point very well. The much higher contribution from





TRA at this sampling point (38.6 ng g$^{-1}$) (Figure 5) is attributed to emissions from Southern Russia (e.g., Tomsk), where all the main cities in Russia are located. Another large fraction of TRA emissions comes from Central and Eastern Europe (see also in http://niflheim.nilu.no/NikolaosPY/SnowBC_141516.py). Similar to TRA, emissions from DOM were mostly transported to Khanty-Masiysk from Central and Eastern Europe, as well as from Turkey contributing 36.6 ng g$^{-1}$ (Figure 5). As previously mentioned, the sampling point where the highest EC concentration was measured is located inside the largest gas flaring region of Russia. In addition, the corresponding emission sensitivity maps showed that the air was coming from south passing directly through this high emission region making FLR emissions the highest contributing source (88.8 ng g$^{-1}$) (Figure 5).

## 4 Discussion

### 4.1 Cross validation of modelled BC concentrations with public datasets

In this section, we present an effort to further validate our model calculations of BC concentrations in snow. For this purpose, BC concentrations in snow were adopted from Doherty et al. (2010) and FLEXPART BC concentrations were simulated as described in section 2.4. Samples were collected in Alaska, Canada, Greenland, Svalbard, Norway, Russia, and the Arctic Ocean during 2005–2009, on tundra, glaciers, ice caps, sea ice, frozen lakes, and in boreal forests. Snow was collected mostly in spring, when the combination of snow cover and exposure to sunlight is at maximum and before the snow had started to melt. Samples of melting snow collected in the summer of 2008 from Greenland and from Tromsø, Norway, were removed from the study, as we have no knowledge about the depth of the melt layer and effects of the percolation of meltwater through the snowpack. All samples were collected away from local sources of pollution. In many locations (Canadian Arctic, Russia, Greenland, Tromsø and Ny-Ålesund) samples were gathered at different depths throughout the snowpack, giving information on the seasonal evolution of BC concentrations as the snow accumulated (and/or sublimated) throughout the winter. In these cases only the surface BC was taken into account. The snow was melted and filtered, and the filters were analysed in a specially designed spectrophotometer system to infer the concentration of BC (for more information see Doherty et al., 2010).

A comparison of measured and modelled BC concentrations in snow is depicted in Figure S 2. The model captures snow BC concentrations effectively in most of the Arctic regions except for the Canadian Arctic, where the modelled concentrations of snow in 2007





were significantly higher. Samples from the same region in other years showed good
agreement with modelled values. The model generally tends to underestimate deposition with
a MFB of -51%, similar to our finding for the new Russian measurements. The RMSE was
estimated to be 52 ng g$^{-1}$, which is acceptable considering that the variation of snow
concentrations in the dataset ranged from 0.3 to 783 ng g$^{-1}$. The highest measured
concentrations of snow BC were observed in Russia, where the model showed a good spatial
agreement. For instance, the highest values were obtained in Western Siberia, close to the gas
flaring regions of the Nenets/Komi oblast, as well as in southeastern and northeastern Russia,
where air masses were arriving from high emitting sources in southeastern Asia. Lower biases
in modelled BC concentrations were observed in northern Siberia with the exception of a few
samples at the coasts of the Kara Sea and northeastern Siberia. Furthermore, biased BC
concentrations were also observed in Greenland and northern Canada. In Western Siberia, BC
in snow presented in Doherty et al. (2010) between 2005–2009 was 101±153 ng g$^{-1}$ on
average, which is very close to the average value of measured EC obtained from the sampling
2014–2016 campaigns (83±37 ng g$^{-1}$).
Moreover, our model was also compared with snow samples collected in a recent
campaign presented in Macdonald et al. (2016). These snow samples were collected at the
Global Atmosphere Watch Observatory at Alert, Nunavut, from September 14$^{th}$, 2014 to June
1$^{st}$, 2015. Alert is a remote outpost in the Canadian high Arctic, at the northern coast of
Ellesmere Island (82°27' N, 62°30' W), with a small transient population of research and
military personnel. Sampling details and analytical methodologies used for the analysis of BC
can be found in Macdonald et al. (2016). BC concentrations in FLEXPART were simulated as
in all previous analyses described in this paper (see section 2.4.). Timeseries of simulated and
measured BC are depicted in Figure S 3 for the whole sampling period. As before, the quite
high correlation coefficient ($R$) of 0.63 indicates that our model captures the temporal
variation of the measured BC in snow. The RMSE was estimated to be almost 63 ng g$^{-1}$, a
relatively high value. The MFB of 47% indicates a strong overestimation of snow
concentrations, although in many samples the opposite was also observed (Figure S 3). This is
in contrast to the previous data sets discussed, for which the model underestimated.
We have further tried to further analyse the origin of the aforementioned
overestimations in the Canadian Arctic in both datasets (Doherty et al., 2010; Macdonald et
al., 2017), as they are shown to be rather systematic. For this reason, we have calculated the
average footprint emission sensitivities and the average BC contribution from the major





sources in ECLIPSE for the 2007 snow samples in the Canada Arctic and for Alert samples that were three or more times higher than the observations, in order to locate the simulated overestimations (Figure 6).

Regarding the model overestimation for the 2007 samples, the average footprint emission sensitivity showed that the air was coming from continental regions of Canada with a smaller contribution from Scandinavia (Figure 6). The highest emission sources for these samples were TRA and DOM that contributed almost 80% to the snow concentrations, whereas forest fires were less important at the time of sampling. Two hot spot areas were identified, one along the borders of Canada with USA and another in southeastern Asia of smaller intensity. A similar emission sensitivity was obtained for the same area of the Canadian Arctic in 2009 only slightly shifted to the north, whereas modelled concentrations were in very good agreement with observations (Figure S 2). This shows that the model overestimation for the 2007 samples is likely attributed to an overestimation of TRA and DOM sources in North America in ECLIPSE for 2007. For the Alert samples that the model strongly overestimated BC, the major sources were TRA and FLR, which contributed 55%, and BB which contributed about 7 ng g$^{-1}$ on average (Figure 6). Anthropogenic BC arriving from Europe and Russia has been previously shown to be important for Alert air concentrations (Sharma et al., 2013). The model overestimation of BC in snow samples at Alert needs further investigation. It is likely that it originates from anthropogenic emissions in northwestern America or in Europe, because forest fires in Canada and Russia, although important for Alert (e.g., Qi et al., 2017), were not significant in the present comparison.

## 4.2 Model deviation from snow EC measurements and region–specific contribution of sources

It was already shown that, on average, measured concentrations of EC in snow in northwestern European Russia and Western Siberia were underestimated in FLEXPART (Figure 2). This was confirmed by the calculated fractional bias (see section 3.2), the spatial distribution of which is shown already in Figure S 1. To examine whether this underestimation was due to missing emission sources or errors in modelled transport and deposition, we have calculated the average footprint emission sensitivity for those sampling points, for which FLEXPART strongly ($FB < -100\%$) and slightly ($-100\% < FB < 0\%$) underestimated the observed values. The average footprint emission sensitivities are shown in Figure 7 together with the locations of the active fires of the last two months until collection



of snow samples adopted from MODIS (Moderate Resolution Imaging Spectroradiometer) (Giglio et al., 2003) and the gas flaring facilities from the Global Gas Flaring Reduction Partnership (GGFR) (http://www.worldbank.org/en/programs/gasflaringreduction).

When the model strongly underestimated the measured EC ($FB < -100\%$), the average footprint emission sensitivity showed the highest values over the Yamal Peninsula and the agglomeration of many gas flares in Khanty-Mansiysk (Figure 7). This might confirm the finding of Huang et al. (2014) that gas flaring emissions in the ECLIPSE inventory, while very high, are still underestimated. According to a related study, Russia contributes 57% to the global BC emissions from gas flaring (Huang and Fu, 2016). Underestimation of modelled atmospheric concentrations compared to observations from the Barents and Kara Seas was recently also reported by Popovicheva et al. (2017), although the underestimation was relatively small.

When the model showed a moderate underestimation of EC concentrations in snow ($-100\% < FB < 0\%$), the emission sensitivity was high near Arkhangelsk and over Scandinavia (Figure 7). BC emissions in Scandinavia are considered relatively low in most inventories and contribute no more than 6.5% to the global emissions in ACCMIP (Aerosol Chemistry Climate Model Intercomparison Project) (Lamarque et al., 2013), 6.2% in EDGARv4.2 (Emission Database for Global Atmospheric Research) (Olivier et al., 2005), 2.1% in MACCity (Monitoring Atmospheric Composition & Climate / megaCITY - Zoom for the ENvironment) (Hollingsworth et al., 2008; Stein et al., 2012) and 3.3% in ECLIPSE (Klimont et al., 2016). The highest emission sensitivity was found over northwestern Russia though (Figure 7), where Murmansk is located. Pollution in Murmansk can be high due to emissions from local industry, mining, heating and transport (Law and Stohl, 2007). Another potential source region was Nenets/Komi area and Western Kazakhstan, where a few other flaring facilities are located (Figure 7).

Figure 7 shows that the underestimation of observed EC concentrations in snow strongly depends on the region, where samples are collected. In Western Siberia, the underestimation was larger than in northwestern European Russia. For this reason, we have computed the average region–specific emission sensitivities and the average region–specific contribution from the major polluting sources of ECLIPSE. We distinguish between three regions, northwestern European Russia, Western Siberia (north of 62 °N) and Western Siberia (south of 62 °N) (Figure S 4 – S 6). For the samples collected in northwestern European



Russia (Figure S 4), an average contribution of 21.6 ng g$^{-1}$ from all sources was estimated
mainly originating mainly from TRA (7.7 ng g$^{-1}$) and DOM (10.4 ng g$^{-1}$) sources in Finland.
The contribution from BB and FLR emissions was insignificant, whereas the rest of the
ECLIPSE sources were negligible (IND, ENE, WST). For the samples collected at high
latitudes in Western Siberia (Figure S 5), the average contribution from all sources was more
than 4 times higher (86 ng g$^{-1}$) than those observed in northwestern European Russia. FLR
emissions accounted for 40% of the total reflecting the proximity to the main flaring facilities
of Russia. Another 24% of the average contribution was attributed to TRA activities in
Europe and southeastern Russia that affect the northern part of Western Siberia, although they
are rather remote. Finally, DOM emissions in Eastern Europe also contributed another 28%.
Finally, for the samples that were collected in the southern part of the Western Siberia (Figure
S 6) an average contribution of 47.4 ng g$^{-1}$ was estimated from all sources included in
ECLIPSE. Again, the highest contributing sub-categories were TRA and DOM, whereas FLR
appeared to contribute rather insignificantly considering that the sampling area is close to
Khanty-Mansiysk flaring region.
Overall, the region–specific analysis of the sources contributing to modelled BC in
snow showed that the DOM, FLR and/or TRA sources might explain the model
underestimation in high Arctic. However, in the most recent assessments of BC of the higher
Arctic (Popovicheva et al., 2017; Winiger et al., 2017), it was shown that ECLIPSE captures
levels of BC quite well, whereas FLR emissions might have a smaller impact in the Central
Siberian Arctic (Tiksi) than previously estimated. Surprisingly, the average contribution from
BB in lower latitudes was extremely low in all Western Siberia (Figure S 5 and S 6), despite
the fact that sampling took place in springtime, where BB becomes important. Evangeliou et
al. (2016) reported that using a different dataset, that is based on the same approach as GFED,
but includes updated emission factors for Eurasia, surface concentrations of BC in the Arctic
stations can be substantially higher. This shows the need for further investigation of BC
sources in the Russian Arctic.

## 5  Conclusions

We have analysed snow samples collected in Western Siberia and northwestern
European Russia during 2014, 2015 and 2016 with respect to EC. This region is of major
interest due to its large uncertainty in BC emissions and because it is located in the main



transport route of BC to the Arctic. An effort to constrain the sources that contribute to the measured concentrations of BC in snow was made using the LPDM FLEXPART (version 10).

The observed EC levels in snow varied widely within and between regions (3–219 ng g$^{-1}$ for 2014, 46–175 ng g$^{-1}$ in 2015 and 7–161 ng g$^{-1}$ in 2016), and are in the upper range of previously reported concentrations of EC and BC in snow in the Arctic region. However, the observed levels presented here appear typical for Western Siberia, which is subject to high domestic Russian emissions, as well as to transport from distant European ones.

The snow BC concentrations predicted by the model are in good agreement with EC observations over Western Siberia and northwestern European Russia ($R = 0.5 - 0.8$). However, the calculated MFB values (-48% to -27%) showed that the model systematically underestimated observations in Russia. This underestimation strongly depended on the region where the samples were collected. In northwestern European Russia, the main contributing sources were TRA and DOM mainly from adjacent regions in Finland. TRA and DOM contributed double to snow BC sampled at low latitudes of Western Siberia (<60°N), with the majority of the emissions to originate from highly populated centres in Central Europe. Finally, in higher latitudes of Western Siberia (>60°N), snow BC concentrations were further increased mainly due to FLR emissions from facilities located close to the snow sampling points.

The modelled BC concentrations in snow were further investigated using two independent public measurement datasets that include samples from all over the Arctic for the period 2005 to 2009 and from Alert in 2014 and 2015. The model captured levels of BC quite effectively despite the large variation in measured concentrations. An exception was observed in North America in spring 2007 and in Alert observatory in late winter – early spring 2015. In both cases, the major sources were along the Canadian borders with USA and in Western Europe. Considering that similar deviations were not observed in the area in samples collected during other years, it is likely that some of the prevailing sources of BC there show strong temporal variability in their emissions, and this is not taken into account in ECLIPSE. Overall, previously reported measurements of snow BC in Western Siberia and northwestern European Russia were 101±153 ng g$^{-1}$ on average, which is about 20% higher than the EC measurements presented here (83±37 ng g$^{-1}$).

*Data availability.* All data used for the present publication can be obtained from the corresponding author upon request.





*Competing interests*. The authors declare that they have no conflict of interest.

*Acknowledgements*. We would like to acknowledge the project entitled "Emissions of Short-Lived Climate Forcers near and in the Arctic (SLICFONIA)", which was funded by the NORRUSS research program of the Research Council of Norway (Project ID: 233642) and the Russian Fund for Basic Research (project No. 15-05-08374) for funding snow sampling in the White Sea catchment area. We also thank Sergey Belorukov, Andrey Boev, Anton Bulokhov, Victor Drozdov, Sergey Kirpotin, Ivan Kritzkov, Rinat Manasypov, Ivan Semenyuk, and Alexander Yakovlev for helping during the three expeditions and Academician Alexander P. Lisitzin for his valuable recommendations. O. S. Pokrovsky and S. N. Vorobiev acknowledge support from BIO-GEO-CLIM grant No 14.B25.31.0001 for sampling in Western Siberia. Acknowledgements are also owed to IIASA (especially Chris Heyes and Zig Klimont) for providing the BC emission dataset. Computational and storage resources for the FLEXPART simulations have been provided by NOTUR (NN9419K) and NORSTORE (NS9419K). All plots from FLEXPART simulations have been included in an interactive website for fast visualization (http://niflheim.nilu.no/NikolaosPY/SnowBC_141516.py). All results can be accessed upon request to the corresponding author of this manuscript.

*Author Contributions*. N. Evangeliou designed and performed the modelling experiments and wrote the paper. V. P. Shevchenko organised and performed the sampling of EC, K.-E. Yttri performed all the TOA of the snow samples. S. Eckhardt modified FLEXPART model for the calculation of footprint emission sensitivities for deposited mass. E. Sollum wrote an algorithm that computes the starting date of the FLEXPART releases based on the water equivalent volume from ECMWF. O. S. Pokrovsky, V. O. Kobelev, V. B. Korobov, A. A. Lobanov, D. P. Starodymova and S. N. Vorobiev assisted the sampling campaigns in Western Siberia and northwestern European Russia during 2014–2016. R. L. Thompson and A. Stohl supervised the study and wrote parts of the paper.

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




**FIGURE CAPTIONS FOR MANUSCRIPT**

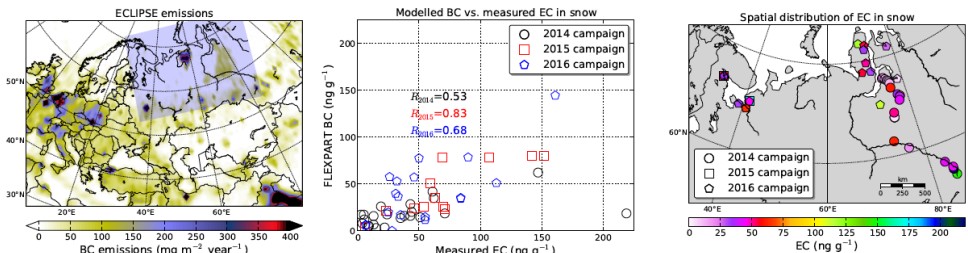


**Figure 1.** Left: Total emissions of BC from the ECLIPSE emission inventory (Klimont et al.,
2016). The blue shade shows the area of interest that is zoomed on the right. Middle:
Comparison of modelled BC concentrations in snow with measured EC concentrations. Right:
Spatial distribution of EC in snow measured by thermal optical analysis (TOA) of filtered
snow samples from northwestern European Russia and Western Siberia in spring–time 2014,
2015 and 2016.








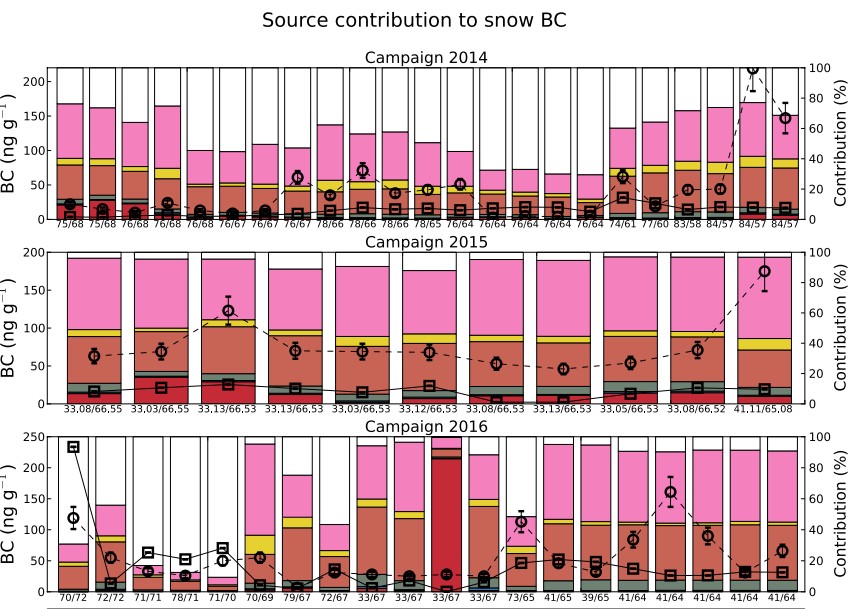


**Figure 2.** Contribution from the various emission categories considered in the ECLIPSE and GFED inventories (BB stands for biomass burning, WST for waste burning, IND for industrial combustion and processing, TRA for surface transportation, ENE for emissions from energy conversion, and extraction, DOM for residential and commercial combustion, and FLR for gas flaring) to simulated BC concentrations in snow during the 2014, 2015 and 2016 campaigns in Western Siberia and northwestern European Russia. Bars show the relative source contribution (0 –100%, right axis) and are sorted, from left to right, from the northernmost to the southernmost measurement location (coordinates are reported on the bottom as longitude/latitude). Measured EC concentrations in snow are reported with open circles, whereas modelled BC is shown with open rectangles (left axis).



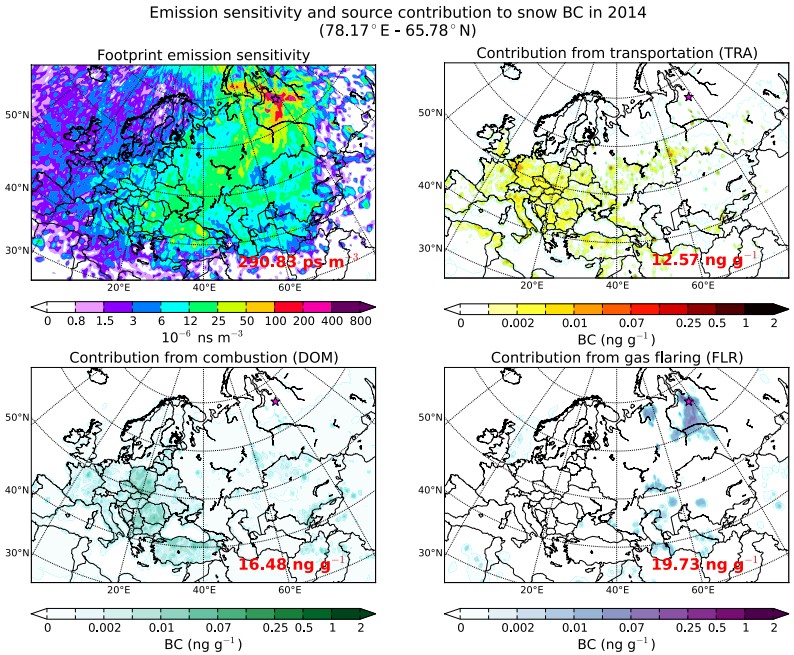

790

**Figure 3.** FLEXPART emission sensitivity (top left) and contribution from transportation (TRA, top right), residential and commercial combustion (DOM, bottom left) and gas flaring (FLR, top right) to the maximum measured concentration of snow EC recorded along the transect from Tomsk to Yamal Peninsula in Western Siberia during the campaign of 2014.





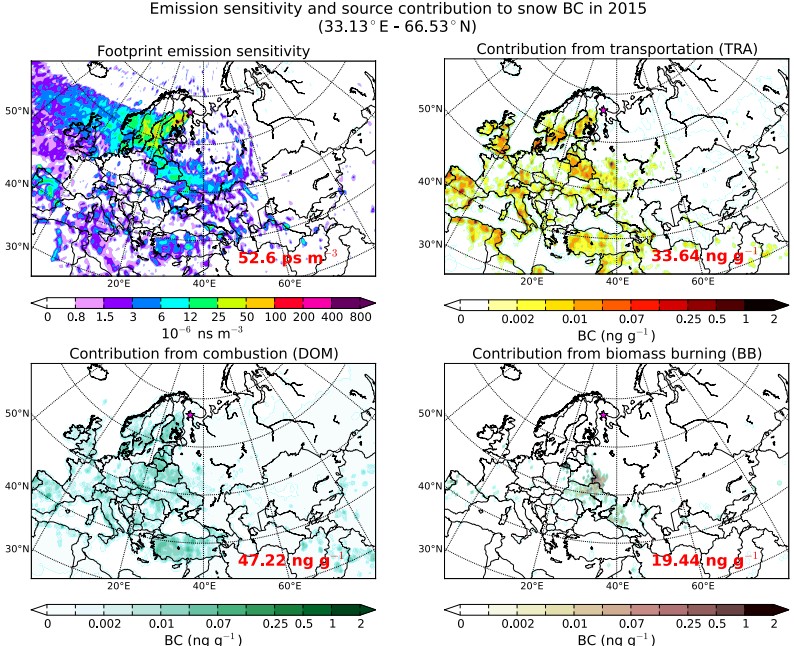


**Figure 4.** FLEXPART emission sensitivity (top left) and contribution from transportation
(TRA, top right), residential and commercial combustion (DOM, bottom left) and gas flaring
(FLR, top right) to the maximum measured concentration of snow EC recorded in
northwestern European Russia (Kindo Peninsula and Arkhangelsk region) during the
campaign of 2015.





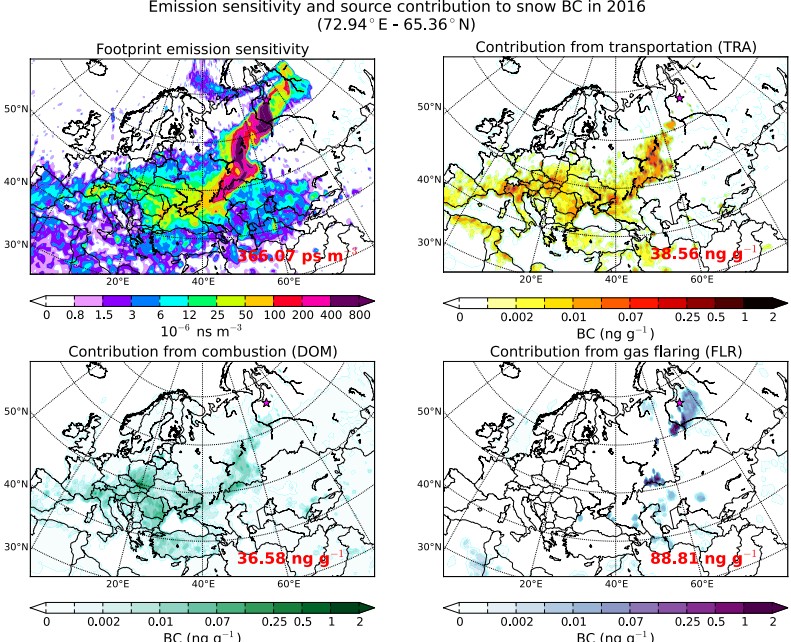


**Figure 5.** FLEXPART emission sensitivity (top left) and contribution from transportation (TRA, top right), residential and commercial combustion (DOM, bottom left) and gas flaring (FLR, top right) to the maximum measured concentration of snow EC recorded in Kindo Peninsula, Arkhangelsk and Yamal Peninsula (northwestern European Russia, Western Siberia) during the campaign of 2016.





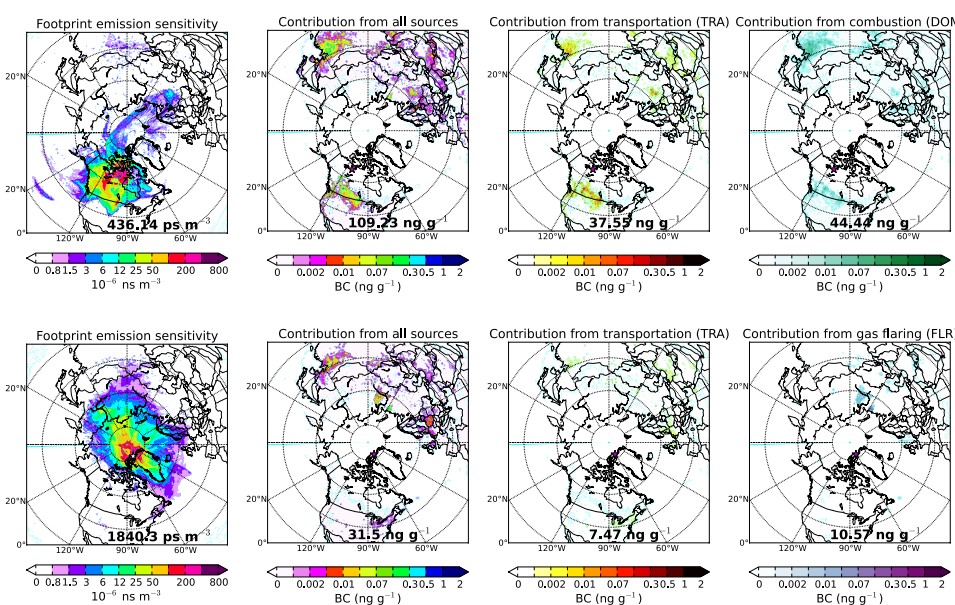

807

**Figure 6.** Top row: Footprint emission sensitivity and major contribution from all sources, TRA and DOM averaged for the samples that showed overestimated modelled concentrations of BC in 2007 (Doherty et al., 2010). Bottom row: Footprint emission sensitivity and contribution from all sources, TRA and FLR for the samples collected in Alert (Macdonald et al., 2017) that model overestimated by more than three times.







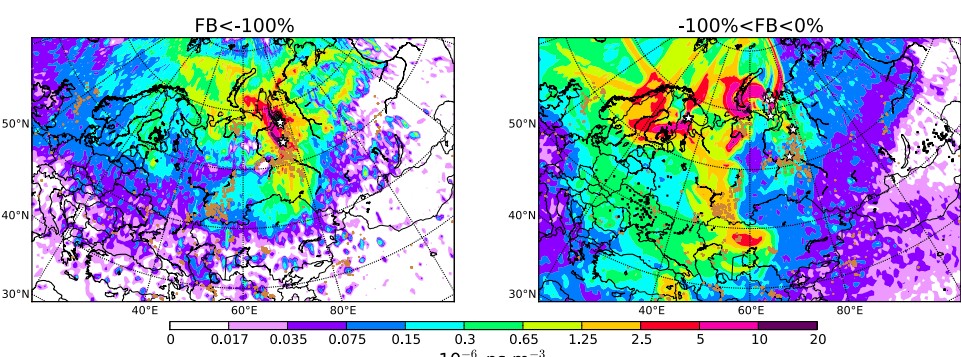


**Figure 7.** Footprint emission sensitivity from FLEXPART averaged for the sampling points
where the model underestimated observations significantly ($FB < -100\%$) and less
significantly ($-100\% < FB < 0\%$). Black squares show the locations of active fires
detected by MODIS (Moderate Resolution Imaging Spectroradiometer) (Giglio et al., 2003).
Brown dots show the location of gas flaring sites from the Global Gas Flaring Reduction
Partnership (GGFR) (http://www.worldbank.org/en/programs/gasflaringreduction).