# Peer review of "Origin of elemental carbon in snow from Western Siberia"

_Atmospheric Chemistry and Physics, 2017_

## Referee Comment (RC1) · Anonymous Referee #1 · 8 Jul 2017

General Comments:

I have no show-stopping issues with the analysis presented or the paper. The authors present a fairly comprehensive analysis of EC from snow samples collected across northern Russia, compare them to modeled values, and do a source apportionment analysis using FLEXPART in a new mode that allows running backtrajectories that track deposited mass, rather than ambient atmospheric concentrations. They also compare their EC concentrations to those from other measurements around the Arctic, and they test modeled EC against these concentrations from other studies.

[Figure]
* * *
Interactive
comment

The data set and analysis presented are useful and the paper should be published once the issues raised below are addressed.

Specific Comments:

Very minor editing for English would be good. (e.g. "a component of the fine particulate matter" "a component of fine particulate matter"; "further tried to futher analyze"; "TRA and DOM contributed double to snow BC sampled at low latitudes"...)

The sampling dates varied from early Feb to late April. When the samples were collected could influence the results in two ways that are not sufficiently discussed: 1) Biomass burning (wildfires) in northern Eurasia can become significant in March to April. The source apportionment (Fig 2) shows a very small role of wildfires, but there is some influence in some of the northern samples in 2015 and a significant role in one of the samples in 2016. It might be useful to indicate in Fig 2 (perhaps above each bar?) what date the samples were collected. 2) It seems possible there might have been some surface melting of the snow before sampling. If this is the case, surface concentrations could be elevated due to consolidation of BC at the snow surface, rather than due to increased deposition. Was there any effort made to determine whether the snow might have experienced melt at some point prior to being sampled? Either way, this should be noted.

Hegg et al. found that biomass burning constituted a significant fraction of BC in snow from their northern Russia samples, in contrast to what you found here (i.e. see Fig 2). Hegg et al. could not distinguish between wildfire emission and domestic wood-burning emissions, so one possible explanation is that a significant fraction of the DOM (domestic burning) category in this study is wood burning. This would bring the source attribution of Hegg et al. and that given here in better agreement. It would be very useful if you could state what is included in the DOM emissions category; whether or not for this region a significant fraction of the DOM emissions are from wood burning; and to compare your source apportionment results to that of Hegg et al.

Pg 2, lines 42-44. "Modelled BC was in good agreement (ðİŚĚ = 0.53 − 0.83) with mea-sured EC. However, a systematic region–specific model underestimation was found." The wording here needs editing. First, R is an measure of correlation, not agreement. R could be 1.0, but if the two differ by a factor of 2 there is hardly "good agreement". Second, an R of 0.53 means R-squared of 0.28, which is not a very high correlation coefficient. I would say they were moderately correlated, and the measured values were higher than the modeled values (by, e.g. "on average, XX%").

Pg. 9, lines 246-250: Same comment as made above re: the text in the Abstract around "good agreement", and confounding "correlation" and "agreement". As discussed in the text that follows, there was often significant bias in the modeled values relative to the measured values!

Pgs. 12-13 and Figure S2 discussion of comparison of FLEXPART and Doherty et al. (2010) results: First, Figure S2 would be more useful if it showed the locations of the samples compared in a map and then the actual comparison in an x-y correlation plot. Trying to compare the two maps as given is not very useful, given the large range in concentrations. In an x-y plot, locations in different regions could be given different symbols, corresponding to the regional comparisons (e.g. Canadian Arctic, Western Siberia) as discussed in the text. Second, again, the text significantly over-states the level of agreement. In this case R is 0.24 (R2<0.06 – i.e. the model only captures <6% of measured variability - !), and there is a 50% bias in the concentrations, on average.

Pg. 13, line 397: Again, R of 0.63 (R2 of 0.29) is not "quite high"

Pg. 13-14: Hegg et al. (2010, ACP) presents a source attribution of the BC in Arctic Canada snow measured by Doherty et al. (2010). It would be good to incorporate these results in the discussion here. Not doing so seems like an omission.

Pg. 17, lines 521-522: "The model captured levels of BC quite effectively despite the large variation in measured concentrations." Again, I disagree with this very optimistic statement of the results of the comparison.

Smaller comments/corrections:

Pg 2, lines 45 and 47: The use of >-100% and <-100% is a bit ambiguous. ">-100%" could be read as more than a factor of 2 difference, and "<-100%" as less than a factor of 2 difference. I'd suggest rewording for better clarity.

Pg. 3 lines 73-75: "Sea ice has a much higher albedo ($\approx$0.5–0.7) compared to the surrounding ocean ($\approx$0.06), thus BC deposited on sea ice reduces the heat uptake of the ocean." I understand what you're trying to say here, but as written it's not accurate: BC deposited on ice does not reduce the heat uptake of the ocean – the presence of sea ice does. BC deposited on ice lowers its albedo, increases heat uptake by sea ice, accelerates its melt, and therefore decreases surface albedo both directly and indirectly.

pg 4, lines 111-112: After discussing (correctly!) that BC/EC are operationally defined it's stated that "In the present study, EC measurement data from three campaigns are compared to simulation results" – without stating what measurement method is used!

Pg 5, lines 133: It is well known that quartz-fiber filters can have low and highly variable capture efficiency for particles in liquid samples. Was capture efficiency tested/measured? If not, at a minimum this potential source of bias needs to be acknowledged. Hopefully, some tests were done. (As an example, Hadley et al., 2008, Env Sci Tech found that to get high filter capture efficiency they had to run the samples through 3 stacked filters$\sim$)

Pg. 6. Line 161: I would reword "driven with 3-hourly" to "3-hour resolution"

Pg. 7, lines 201-203: "Assumed aerodynamic mean diameter and logarithmic standard deviation are used by FLEXPART's dry deposition scheme, which is based on the resistance analogy. . ." The assumed size for BC (0.25 microns) is reasonable. However, the deposition rate should be driven by the size of the particles *containing* the BC. It is very unlikely that the BC in the atmosphere was externally mixed with other aerosol

components; much more likely is that multiple components were internally mixed in larger particles. This would affect dry deposition rates based on resistance.

Pg. 10, pg 287-289: Doherty et al. (2010) specifically measured BC in snow in northern Russia, including western Russia. It's odd not to note this, and to not compare your results directly with theirs from a similar region. Also it's odd to only state that concentrations were "up to 800ng/g", rather than discussing more representative results from their analysis.

Pg. 26, Figure 1 caption: It might be good to remind the reader in the caption that the ECLIPSE emissions don't include wildfire emissions.

Pg. 27, Figure 2 caption: Some rewording/re-parsing of the (very long!!!) first sentence of this caption would make it much more readable. . .

Figure 1: Right-most panel, showing spatial distribution of EC concentrations. I found the color-scale used here not very intuitive. It might be better to go from, e.g., dark blue for low values to bright red for high values.

Figures 3-5: I found the little red stars indicating sampling location difficult to find. I'd suggest making this symbol larger.

---

## Referee Comment (RC2) · O. ÂăG. FAWOLE (Referee) · 7 Dec 2017

O. ÂăG. FAWOLE (Referee)

gofawole@oauife.edu.ng

Origin of elemental carbon in snow from Western Siberia and northwestern European Russia during winter-spring 2014, 2015 and 2016 Author: Nikolaos Evangeliou et al.

General Comments 1. The manuscript present results of elemental carbon (EC) concentration in snow samples collected at various locations in the regions of Western Siberia and northwestern European Russia. It also presented output of LDMP FLEX-PART model which was used to identify the major sources which contributed to the BC

concentrations measured in the snow samples. 2. The manuscript needs adequate grammar editing. The construct and flow of some of the sentences need to be reconstructed. The use of opening phrasal nouns and adjectives are often out of place. 3. A major deficiency of the manuscript is the labelling of the individual plots in the Figures. Figures 1 - 7 should be labelled (a), (b), (c), (d) and so on as appropriate. This will make your discussion of the figure easier. 4. The sentences in lines 41 and 52 are contradicting each other. 5. Section 3 (Results) of the manuscript presented both results and discussion of the various analysis rather than the results. Detailed discussions of the results should be in section 4 (Discussion) along with the cross validation of the model and model deviation. 6. Under sample collection, it is necessary to highlight the number of samples collected at each site and the total number for each year. Also, provide a separate figure of the sampling sites preferably a map. When making reference to the sampling site, you refer the readers to Figure 1 which did not show the sampling sites explicitly. 7. For the concentration of EC in snow (section 3.1), you could report the percentile (upper and lower) instead of the standard deviation. 8. Some of the data compressed into section 3.1 could be better understood by the reader if they are presented in tabular form. 9. For the cross validation (section 4.1), it will be better to state explicitly that you used FLEXPART to simulate BC concentration for Doherty and Macdonald's dataset. 10. You have used different reference format for the manuscript and supplementary materials. 11. In line 142, what is the performance compared with? 12. In section 2.3, what do you mean by carbonate ($CO_2$-3)-carbon? Do you mean carbonate ($CO_2$-3)?

Specific Comments: 1. Line 38: Why did you refer to the recently developed algorithm as feature? I think it should be recently developed algorithm o routine. 2. Line 39: backwards should be backward 3. Line 57: most strongly should be strongest. Delete 'the'. That part of the sentence should read "component of atmospheric aerosol." 4. Line 65: should read "BC is important on a global perspective because of its . . . . . . . . . . . .." 5. Lines 65-66: provide a reference for the opening sentence. 6. Line 66: should read "As a component of fine particulate matter . . . . . . . . . . . . . . . .." 7. Line 69: should read

"……………... it absorbs radiation and accelerates melting of the ice. 8. Lines 91-93: The references cited here are not properly cited. The last part after the unnecessary full stop should not be in a bracket. 9. Line 101: are major sources of what in the area? 10. Lines 104 -105: The references in the bracket should be preceded by 'for example' since the references are just examples of articles that have used EC and BC. 11. Lines 107 – 110: The statement "consequently, BC …………………….the world" added no substantial meaning to the discussion here. Hence, I suggest you expunge it. 12. Line 110: should read "……………………..BC should be used quan-titatively" 13. The statement "In the present study, …………………." should start a new paragraph. 14. Line 119: re-cast the statement beginning from "near the port". The near ……… near in the last part of the statement makes it ambiguous. 15. Line 120: Is Kindo Peninsula in Arkhangelsk or Arkhangelsk is a sampling site on its own? 16. Line 121: should read "……………….. Pollution levels in these areas have been partly attributed to urban and gas flaring sources." 17. Line 125: should read "……………….. to minimise the direct influence from ……." 18. Line 126: should read "…………….. information about sample collection such ….." 19. Line 127: should read "……………. and the depth at which snow was sampled …" 20. Line 129: should read "Sampling was perform with a metal-free technique using pre-cleaned …………………………' 21. Line 130: should read "……… polyethylene bags which had been ……………" 22. Line 131: should read "……….. 1M HCl and rinsed with abundant deionised ultrapure water in the ………………….." 23. Line 133: should read " ………….. filtered through 47 mm quartz fibre filters. The filters were dried at 60-70 oC …………………………..'' 24. Line 138-139: should read "Ele-mental carbon content of the filters were measured …………….. (TOA) using the sunset laboratory …………..'' 25. Line 142: should read "Performance of the OC/EC instrument is regularly ………'' 26. Line 143-144: Recast leaving out the slash af-ter (EMEP). 27. Line 148-149: should read "The carbonate content of filtrate on the filters was measured by TOA after thermal-oxidative ………………..'' 28. Line 150: should read "A punch of 1.5 cm2 ……………………" 29. Line 152-154: Re-cast this

sentence. Do you mean section 2.2 or chapter 2.2? 30. Line 156 and 157: 'evolves' should be 'evolved' 31. Line 158: should read "Applying this correction, EC values were . . . . . . . . . . ." 32. Line 160: Give the full meaning of LPDM at first use. 33. Lines 163-165: Re-cast to read "The ECMWF data has 137 vertical data and a horizontal resolution of 1 x 1 for 2014 and 2015 simulation, and 0.5 x 0.5 for 2016" 34. Line 188: mass per unit area you mean 35. Could you re-cast this sentence? 36. Line 198-200: What are the rationale/references for these assumptions? Any similar assumption in literature? 37. Line 214: this should be Figure 1(c). 38. Line 216: Like I stated in the general comment, you could report the 25th and 75th percentile or 10th and 90th percentile. 39. Line 221: should read ". . . . . . ..the snow samples for 2014, EC concentrations . . ..." 40. Line 228: should read ". . .. . . . . . . . . .(on the White sea coast) showed high . . . . . .." 41. Line 232-240: Re-cast the five sentences in these lines. 42. Line 239: should read " . . .. . .. . . .. . . . .Tomsk and Yamal, EC concentration was highly . . . .. . .. . .. . . . . . .. . . . . . . .." 43. Line 244: Should read ". . . .. . . . . ...measured EC concentrations in the snow samples . . . . .. . . . . . . . . . . .." 44. Line 246: A scatter plot of what? Figure 1 should be Figure 1(b). 45. Line 247: should read ". . . . . . . . . . . . agreement and good correlation . . . . . . . . . . 46. Line 258: The sentence "The MFB of the . . . . . . . . . . . . . . .. was -42%" is somehow isolated. What inference can be drawn from the fact that MFB is -42%. 47. Line 264: should read "For 2016, FB values . . . . . . . . . . . . . . . . . .. . . . . . . .. show another set of underestimation. 48. Line 266: 12 out of 19 what? Samples? 49. Line 266: Should read "19 samples. For the remaining 7 samples, the model . . . . . . .." 50. Line 267 should read ". . . . . . . . . . . . . . .. The root mean square error . . . . . ." 51. Line 268-269: Please, re-frame this sentence. The sentence, as it stands, is ambiguous. I guess it should read "The RMSE is frequently used to measure . . . . . .." 52. Line 273-275: the sentence is muddled up. What exactly do you want the reader to infer from the two short sentences? 53. Line 276-277: should read ". . . .. reported that the maximum BC concentration measured . . . . . . . .. . . . . . . .." 54. Line 283: should read ". . . . . . . . . . . .Stockholm with a population of about 2 million. 55. Line 287: What do you mean by one order of magnitude?

56. Line 290: should read ".......... Macdonald et al., (2017) reported BC concentrations ranging from ..................... For the samples collected near ..........."
57. Line 295: should read "......ECLIPSE emissions dataset ................." The word 'account' does not work here. Please, choice a different word. 58. Line 297-298: should read "........................ gas flaring (FLR) while biomass burning .............." 59. Line 310-311: the list of cities in the bracket is just too long. Include only the important cities and move the bracket to immediate after 'major Russian cities on line 309. 60. Line 313-316: re-cast this sentence to reflect what you want the reader to understand from the sentence, 61. Line 319: should read "(6%) (see Figure 2).................." 62. Line 320-321: Are these two sources new? Where they not there in 2014? 63. Line 325-326: should read "........ Peninsula whereas FLR emissions ......... were very low due to the long distance of flaring emission sources from the sampling point." 64. Line 327-328: should read ".... also affected BC concentration in snow in northwestern ..........." 65. Line 329: should read " releases in Russia, the miscalculation ........... and their impact in .............." 66. Line 331-332: should read "...........BB emissions, originating mostly from eastern Europe, contributed about .........." 67. Line 336: should read "..... Yamal, DOM, FLR TRA contributed, on the average, 31%, 29% and 27%, respectively (see Figure 2(c))." 68. Line 341: it should be Figure 5(b) if you effect the comment on labelling of individual figures in the plot as suggested in the general comment section 69. Line 353-359: Re-phrase the sentences on these lines stating what exactly you did will the data from Doherty and Macdonald as well as the reasons for the cross validation. 70. Line 374-375: should read" ........... Similar to our finding for the new Russian measurements, the model output, with a MFB of -51%, tends to underestimate deposition." 71. Line 383-384: Are you referring to Doherty data here? If so, state that explicitly. 72. Line 388: Expunge "Moreover" The sentence should read" Our model output was ...with measured BC concentrations in ......" 73. Line 392: 'research' should be 'researcher' 74. Be explicit. Did you do a model run for the period for which Macdonald et al carried out measurements? 75. Line 401: underestimated what? 76. Line 402:

[Figure]

Interactive
comment

should read "Further analysis was carried out to adequately understand . . .." 77. Line 404-408: Re-cast this complex sentence into 2 – 3 simple ones. 78. Line 413: should read ". . . . . . . . . .Two hotspots were . . . . . .." 79. Line 414: should read ". . .. And another, of smaller intensity, in southeastern Asia." 80. Line 415-417: The two simple sentences here are disjointed. 81. Line 419: should read". . . . . . America in ECLIPSE. The Alert samples, for which the model strongly underestimated BC, the major sources . . . . . . . . . . ..". 82. Line 421: Why is 7 ng g-1 not in percentage? 83. Line 422: should read". . . . . . . . . Alert air pollutant concentrations . . . . . . . .." 84. Line 429: should read "It has been shown that average measured . . . . . . . . . .." 85. Line 432: delete 'already.' 86. Line 437: should read". . . . . . . . . . locations of fires that have been active in the last two months before the sample collection. The fire data were adopted from MODIS . . .." 87. Line 439: gas flaring facilities or gas flaring data? 88. Line 443: How do you mean? Around gas flaring facilities? 89. Line 445: should read". . . . . . According to a related study by Huang and Fu (2016), . . . . . . . . .." 90. Line 450: Which model are you referring to here? 91. Line 451-452: These cities/regions are not explicitly labelled in the plots. So that the reader can follow through with the discussions, it is better to include lon/lat of these cities/regions in a bracket. Could you do this for other locations in similar discussion throughout the manuscript at their first mention? 92. Line 452: should be Figure 7(b) 93. Line 458-459: should read". . . . . . northwestern Russia, a region which includes Murmansk. Pollution level in Murmansk could be high due to . . . . . . . . .." 94. Line 462: You have referred to figure 7 severally but these cities are not explicitly shown in Figure 7. 95. Line 467: should read". . . . . . .. polluting sources identified in ECLIPSE dataset." 96. Line 471: should read "to have originated mainly . . . . . . . . . .." 97. Line 472-473: Re-cast this sentence. Insignificant? Negligible? 98. Line 474: delete (Figure S5). It makes no contribution to the sentence. 99. Line 475: should read". . . .. European Russia (Figure S5)" 100. Line 476: should read". . . . . ..of the total contribution, which reflect the proximity of the sampling site to the main flaring facilities in Russia." 101. Line 477-479: could you re-phrase this sentence? 102. Line 480: delete Figure (S6). It adds nothing to the understanding of this sentence, 103. Line

482: Delete 'Again' sub-categories should be 'categories.' 104. Line 483-484: What could be responsible for the insignificant contribution of FLR at this sampling site? Is the site upwind of the flaring facilities? It would be interesting if you could put forward an argument for this insignificant contribution despite the closeness to the flaring facilities. 105. Line 499: : should read"……….... Russia in 2014, 2015 and 2016 …..… EC concentration" 106. Line 501-502: should read"….. to measured BC concentration in snow …." 107. Line 507: : should read"……. Russian emission as well as ……..." 108. Line 515: should read "…………. emissions originating from highly……." 109. Line 525-526: should read"….. Considering the fact that similar ………….observed in samples collected in the area during other years, it is likely … of BC in this region show ………." 110. Line 528: should read"…. previously reported average measurements of BC concentrations in snow in Western …….." 111. Line 529: delete 'on average'. 112. Line 584: Delete one of the 'doi'. 113. Line 623-626: This reference is cited as 2016 in the manuscript (see line 207).

---

## Author Comment (AC1) · 12 Dec 2017

General Comments: I have no show-stopping issues with the analysis presented or the paper. The authors present a fairly comprehensive analysis of EC from snow samples collected across northern Russia, compare them to modeled values, and do a source apportionment analysis using FLEXPART in a new mode that allows running back trajectories that track deposited mass, rather than ambient atmospheric concentrations. They also compare their EC concentrations to those from other measurements around the Arctic, and they test modeled EC against these concentrations from other studies.

[Figure]

The data set and analysis presented are useful and the paper should be published once the issues raised below are addressed.

Response: We would like to acknowledge the reviewer for his very constructive comments on some issues that we had not taken into account before. We believe we have addressed all of his comments.

Specific Comments: Very minor editing for English would be good. (e.g. "a component of the fine particulate matter" "a component of fine particulate matter"; "further tried to futher analyze"; "TRA and DOM contributed double to snow BC sampled at low latitudes". . .).

Response: Corrected.

The sampling dates varied from early Feb to late April. When the samples were collected could influence the results in two ways that are not sufficiently discussed: 1) Biomass burning (wildfires) in northern Eurasia can become significant in March to April. The source apportionment (Fig 2) shows a very small role of wildfires, but there is some influence in some of the northern samples in 2015 and a significant role in one of the samples in 2016. It might be useful to indicate in Fig 2 (perhaps above each bar?) what date the samples were collected. 2) It seems possible there might have been some surface melting of the snow before sampling. If this is the case, surface concentrations could be elevated due to consolidation of BC at the snow surface, rather than due to increased deposition. Was there any effort made to determine whether the snow might have experienced melt at some point prior to being sampled? Either way, this should be noted.

Response: We agree with the reviewer that these reasons could potentially change concentrations of snow EC a lot.

As for biomass burning, we found a small contribution to snow BC in the majority of the samples. However, we believe that this is more or less expected simply because sampling took place in spring-time and it has been previously show using several different approaches and datasets that the hot season of biomass burning in Eurasia is rather summer (see Hao et al. doi:10.5194/gmd-9-4461-2016). We also wanted to put dates on Figure 2. However, due to lack of enough space to put dates in a comprehensive way, we decided to put coordinates and place all the meta-data of the samples in a separate Table that is placed in the Supplements of this article (Table S1).

As regards to the possibility of collection of melting snow, the sapling campaigns in these 3 years were designed such as that the sampling included only fresh snow and NOT melting snow. I now make in more clear in the beginning of section 2.1.

Hegg et al. found that biomass burning constituted a significant fraction of BC in snow from their northern Russia samples, in contrast to what you found here (i.e. see Fig 2). Hegg et al. could not distinguish between wildfire emission and domestic wood-burning emissions, so one possible explanation is that a significant fraction of the DOM (domestic burning) category in this study is wood burning. This would bring the source attribution of Hegg et al. and that given here in better agreement. It would be very useful if you could state what is included in the DOM emissions category; whether or not for this region a significant fraction of the DOM emissions are from wood burning; and to compare your source apportionment results to that of Hegg et al.

Response: Hegg et al. have used a completely different approach to address contribution of different sources to snow BC using a chemical analysis combined with Positive Matrix Factorization. On the contrary, we used a Langrangian Particle Dispersion Model combined with the most updated gridded emissions from ECLIPSEv5 (see methodology in section 2). This means that we use a preset portion of each of the sources that is already know and well documented in ECLIPSE website (see: http://www.iiasa.ac.at/web/home/research/researchPrograms/air/Global_emissions.html) and in Klimont et al. paper (doi: 10.5194/acp-17-8681-2017). In the aforementioned paper, section 3.1 describes all the approaches that used to produce what in the paper we call as DOM sector. Again, to be clear, all the different emission sectors used is ECLIPSE and they were used to generate Figure 2 are constant gridded sources from ECLIPSE. Furthermore, Hegg et al. has treated with his PMF model lots of chemical measurements from 36 samples in high latitudes. If you look at the Supplementary information of his article (http://pubs.acs.org/doi/suppl/10.1021/es803623f/suppl_file/es803623f_si_001.pdf), only 8 samples were collected from the vicinity of Russia (the rest were from Greenland, N. Pole, and N. America) and from completely different regions as we did. Therefore, we do not see how I would compare 2 different things both in terms of methodological and spatiotemporal manner.

Pg 2, lines 42-44. "Modelled BC was in good agreement (Ãř ÌǦIS ÌĄ EËǦ = 0.53 − 0.83) with mea- sured EC. However, a systematic region–specific model underestimation was found." The wording here needs editing. First, R is an measure of correlation, not agreement. R could be 1.0, but if the two differ by a factor of 2 there is hardly "good agreement". Second, an R of 0.53 means R-squared of 0.28, which is not a very high correlation coefficient. I would say they were moderately correlated, and the measured values were higher than the modeled values (by, e.g. "on average, XX%").

Response: Corrected.

Pg. 9, lines 246-250: Same comment as made above re: the text in the Abstract around "good agreement", and confounding "correlation" and "agreement". As discussed in the text that follows, there was often significant bias in the modeled values relative to the measured values!

Response: Corrected.

Pgs. 12-13 and Figure S2 discussion of comparison of FLEXPART and Doherty et al. (2010) results: First, Figure S2 would be more useful if it showed the locations of the samples compared in a map and then the actual comparison in an x-y correlation plot. Trying to compare the two maps as given is not very useful, given the large range in concentrations. In an x-y plot, locations in different regions could be given different

symbols, corresponding to the regional comparisons (e.g. Canadian Arctic, Western Siberia) as discussed in the text. Second, again, the text significantly over-states the level of agreement. In this case R is 0.24 ($R^2 < 0.06$ – i.e. the model only captures <6% of measured variability - !), and there is a 50% bias in the concentrations, on average.

Response: We partly agree with the reviewer in this comment. However, when plotting the data on a x-y correlation plot we end up with Figure 1.

This figure basically shows nothing and no comparison can really be done. This is expected, because the data range within 2 orders of magnitude (0.3 to >400 ng/g) and hence the only way to show them in an x-y plot is to use logarithmic axes like in Figure 2. Here, the modeled results are more centered towards the 1:1 line, but still the plot does not say the overall truth, because the axes are in a log scale. Furthermore, in the last figure, we know nothing about where exactly the model fails to predict measurements, which is not the case when plotting the data on a map.

Because the figure refers to data that were used for supporting validation only and it is only shown in the Supplementary information of this article, we would like to keep the figure as it is. If the reviewer still insists, we are willing to change it in a next step of the reviewing process. The sentences on correlations and agreements have been corrected.

Pg. 13, line 397: Again, R of 0.63 ($R^2$ of 0.29) is not "quite high"

Response: Corrected.

Pg. 13-14: Hegg et al. (2010, ACP) presents a source attribution of the BC in Arctic Canada snow measured by Doherty et al. (2010). It would be good to incorporate these results in the discussion here. Not doing so seems like an omission.

Response: I have tried to incorporate some of the main results at the end of the first paragraph in section 4.1, although I do not see how this study is related to what we try to do here, given that we use the data for validation only and NOT for interpretation,.

Pg. 17, lines 521-522: "The model captured levels of BC quite effectively despite the large variation in measured concentrations." Again, I disagree with this very optimistic statement of the results of the comparison.

Response: Corrected.

Smaller comments/corrections:

Pg 2, lines 45 and 47: The use of >-100% and <-100% is a bit ambiguous. ">-100%" could be read as more than a factor of 2 difference, and "<-100%" as less than a factor of 2 difference. I'd suggest rewording for better clarity.

Response: We agree and have corrected this part.

Pg. 3 lines 73-75: "Sea ice has a much higher albedo ($\approx$0.5–0.7) compared to the surrounding ocean ($\approx$0.06), thus BC deposited on sea ice reduces the heat uptake of the ocean." I understand what you're trying to say here, but as written it's not accurate: BC deposited on ice does not reduce the heat uptake of the ocean – the presence of sea ice does. BC deposited on ice lowers its albedo, increases heat uptake by sea ice, accelerates its melt, and therefore decreases surface albedo both directly and indirectly.

Response: Corrected.

pg 4, lines 111-112: After discussing (correctly!) that BC/EC are operationally defined it's stated that "In the present study, EC measurement data from three campaigns are compared to simulation results" – without stating what measurement method is used!

Response: Corrected.

Pg 5, lines 133: It is well known that quartz-fiber filters can have low and highly variable capture efficiency for particles in liquid samples. Was capture efficiency tested/measured? If not, at a minimum this potential source of bias needs to be acknowledged. Hopefully, some tests were done. (As an example, Hadley et al., 2008,

Env Sci Tech found that to get high filter capture efficiency they had to run the samples through 3 stacked filtersâĹij)

Response: We acknowledge that the collection efficiency of BC in liquid samples by quartz fiber filters can be less than 100%, as reported by Ogren et al. (1993) and Hadley et al. (2008). Differences in collection efficiencies between quartz fiber filters from different manufacturers, and even between batches, can be speculated. Unfortunately, no attempt to estimate the collection efficiency was performed in the present study, and estimating this based on previous studies is speculative. Thus, the results presented should be regarded as conservative estimates. We have included a sentence in the paper to account for this (see line 182-185).

Pg. 6. Line 161: I would reword "driven with 3-hourly" to "3-hour resolution"

Response: Modified.

Pg. 7, lines 201-203: "Assumed aerodynamic mean diameter and logarithmic standard deviation are used by FLEXPART's dry deposition scheme, which is based on the resistance analogy. . ." The assumed size for BC (0.25 microns) is reasonable. However, the deposition rate should be driven by the size of the particles *containing* the BC. It is very unlikely that the BC in the atmosphere was externally mixed with other aerosol components; much more likely is that multiple components were internally mixed in larger particles. This would affect dry deposition rates based on resistance.

Response: BC particles in fresh exhaust are typically found in the 100 nm range or smaller and, in the urban environment, grow relatively quickly to sizes of about 200 nm (e.g., Ning et al., 2013). We agree that this occurs mainly via internal mixing with other types of aerosols. In remote areas, BC is mostly part of the internal aerosol mixture, with typical sizes of around 200 nm (see Freud et al., 2017, for Arctic size distributions). The wet diameters (which determine the physical behavior of the particles such as settling) will be larger than that.

FLEXPART uses a single size distribution for BC aerosols and it does not account for particle growth. Therefore, a size distribution must be chosen that is representative for a broad range of conditions. Our size distribution is not representative for the external mixture of fresh BC particles (which are much smaller) but rather for the internal mixture of aerosols encountered in the Arctic and during most of the time BC resides in the atmosphere. It would not be appropriate to simulate the behavior of BC in fresh exhaust. Thus, while we totally agree with the reviewer about the mixing state of the BC particles, we think our settings are representative of this.

REFERENCES

Freud, E., Krejci, R., Tunved, P., Leaitch, R., Nguyen, Q. T., Massling, A., Skov, H., and Barrie, L.: Pan-Arctic aerosol number size distributions: seasonality and transport patterns, Atmos. Chem. Phys., 17, 8101-8128, https://doi.org/10.5194/acp-17-8101-2017, 2017.

Ning, Z., Chan, K.L., Wong, K.C., Westerdahl, D., Mocnik, G., Zhou, J. H., Cheung, C.S.: Black carbon mass size distributions of diesel exhaust and urban aerosols measured using differential mobility analyzer in tandem with Aethalometer, Atmos. Environ., 80, 31-40, 2013

Pg. 10, pg 287-289: Doherty et al. (2010) specifically measured BC in snow in northern Russia, including western Russia. It's odd not to note this, and to not compare your results directly with theirs from a similar region. Also it's odd to only state that concentrations were "up to 800ng/g", rather than discussing more representative results from their analysis.

Response: Done! We have added a short comparison in section 4.1 about BC and BC measured in samples from the Yamal peninsula. We only give an overview of the levels of concentrations as the information on the metadata is rather poor, the samples were not taken from exactly the same coordinates and they were also collected in different years.

Pg. 26, Figure 1 caption: It might be good to remind the reader in the caption that the ECLIPSE emissions don't include wildfire emissions.

Response: Modified.

Pg. 27, Figure 2 caption: Some rewording/re-parsing of the (very long!!!) first sentence of this caption would make it much more readable. . .

Response: Modified.

Figure 1: Right-most panel, showing spatial distribution of EC concentrations. I found the color-scale used here not very intuitive. It might be better to go from, e.g., dark blue for low values to bright red for high values.

Response: If we do what the reviewer suggests we end up with Figure 3. The figure shows no color variation on the measured EC concentrations. This is because the majority of the measured concentrations were between 0-100 ng/g (blue) and only 3-4 samples above. Therefore, we would like to keep the same colorbar as before in order to show discrete colors for all samples.

Figures 3-5: I found the little red stars indicating sampling location difficult to find. I'd suggest making this symbol larger.

Response: Modified.
* * *
Fig. 1.

Fig. 2.

[Figure]

Fig. 3.

---

## Author Comment (AC2) · 12 Dec 2017

General Comments 1. The manuscript present results of elemental carbon (EC) concentration in snow samples collected at various locations in the regions of Western Siberia and northwestern European Russia. It also presented output of LDMP FLEX-PART model which was used to identify the major sources which contributed to the BC concentrations measured in the snow samples.

Response: We appreciate reviewer for his thorough grammar and syntax editing. In-

deed the manuscript reads much better now and it is clearer in sections that were not previously.

2. The manuscript needs adequate grammar editing. The construct and flow of some of the sentences need to be re- constructed. The use of opening phrasal nouns and adjectives are often out of place.

Response: We have followed reviewer's suggestions. We appreciate reviewer's help for this.

3. A major deficiency of the manuscript is the labelling of the individual plots in the Figures. Figures 1 - 7 should be labelled (a), (b), (c), (d) and so on as appropriate. This will make your discussion of the figure easier.

Response: Corrected.

4. The sentences in lines 41 and 52 are contradicting each other.

Response: Corrected.

5. Section 3 (Results) of the manuscript presented both results and discussion of the various analysis rather than the results. Detailed discussions of the results should be in section 4 (Discussion) along with the cross validation of the model and model deviation.

Response: Generally, we are not keen on presenting measurements or modeling results with numbers only. We also like to trigger deeper on what data represent, otherwise we have a rather boring manuscript. In this sense, we present the results of the snow concentrations of BC in section 3 and the simulations of the LPDM FLEXPART in the same section. In discussions' section, we have a more general discussion of our results, perform an extra validation of the new feature of our model and also try to find potential patterns of areas that our model fails or succeeds to capture measured concentrations. We believe that this structure is appropriate for the presentation of this study. However, if the reviewer insists to restructure the whole manuscript, we would need further, more specific, instructions.

6. Under sample collection, it is necessary to highlight the number of samples collected at each site and the total number for each year. Also, provide a separate figure of the sampling sites preferably a map. When making reference to the sampling site, you refer the readers to Figure 1 which did not show the sampling sites explicitly.

Response: The total number of samples analysed per year in now shown in lines 139-142. We do not really understand what else is needed at this point. Figure 1 (a) shows a map of Europe and a shaded blue area, which is the sampling location. Then we zoom in the highlighted region and present the sampling points in Figure 1 (c). In Figure 1 (c), we use 3 different markers corresponding to each different year and also show different colors that correspond to BC concentrations in a colormap.

7. For the concentration of EC in snow (section 3.1), you could report the percentile (upper and lower) instead of the standard deviation.

Response: The number of samples that we measured during the three campaigns in 2014, 2015 and 2016 was 23. 11 and 20, respectively. We think that the number of samples is very low to present percentiles. On the other hand, presenting minimum and maximum ranges and medians with standard deviations certainly gives an overview of the concentrations level, which is also shown in Fig. 1 (c) and in Fig. 2.

8. Some of the data compressed into section 3.1 could be better understood by the reader if they are presented in tabular form.

Response: Corrected. Please see Table S2 in the Supplementary Information.

9. For the cross validation (section 4.1), it will be better to state explicitly that you used FLEXPART to simulate BC concentration for Doherty and Macdonald's dataset.

Response: Corrected. Please see in line 458 and 525, respectively.

10. You have used different reference format for the manuscript and supplementary materials.

Response: Corrected.

11. In line 142, what is the performance compared with?

Response: Usually, accredited laboratories are obliged to participate in intercomparison exercises. This is a common procedure and there are several references about this. Thus we frequently measure EC in filter samples using the TOA technique for such intercomparisons.

12. In section 2.3, what do you mean by carbonate (CO2-3)-carbon? Do you mean carbonate (CO2-3)?

Response: It is the carbonate content of CO32- which is the issue when it interferes with the OC and EC during TOA, not the oxygen content. The term is commonly used when addressing such issues. We would like to keep this term in the manuscript.

Specific Comments:

1. Line 38: Why did you refer to the recently developed algorithm as feature? I think it should be recently developed algorithm o routine.

Response: We understand that the reviewer is not experienced with the Lagrangian Particle Dispersion Model (LPDM) FLEXPART. So far, the model (or the algorithm is you like) could track atmospheric concentrations back in time. In the development we have done (see Eckhardt et al., 2017), in the same model framework (FLEXPART), we have added the possibility for the user to be able to simulate wet and dry deposition back in time. This is simply done by changing one parameter (details can be seen in Eckhardt et al. paper). This is the reason that we refer to this development as feature to the already existing model framework FLEXPARTv10.

2. Line 39: backwards should be backward

Response: Corrected.

3. Line 57: most strongly should be strongest. Delete 'the'. That part of the sentence

should read "component of atmospheric aerosol."

Response: Corrected.

4. Line 65: should read "BC is important on a global perspective because of its . . .. . .. . .. . ..."

Response: Corrected.

5. Lines 65-66: provide a reference for the opening sentence.

Response: References are given in the next sentences.

6. Line 66: should read "As a component of fine particulate matter . . .. . .. . .. . .. . .. . ..."

Response: Corrected.

7. Line 69: should read ". . .. . .. . .. . .. . ... it absorbs radiation and accelerates melting of the ice.

Response: Corrected.

8. Lines 91-93: The references cited here are not properly cited. The last part after the unnecessary full stop should not be in a bracket.

Response: Corrected.

9. Line 101: are major sources of what in the area?

Response: Corrected.

10. Lines 104 -105: The references in the bracket should be preceded by 'for example' since the references are just examples of articles that have used EC and BC.

Response: Corrected.

11. Lines 107 – 110: The statement "consequently, BC ........................the world"

**[ACPD](about:blank)**
added no substantial meaning to the discussion here. Hence, I suggest you expunge it.

Response: We disagree. This sentence tells a lot about BC and points to the long-term discussion between experimentalists and modellers for the right use of terms BC, eBC, rBC etc. We would like to keep it as it is.

12. Line 110: should read "........................BC should be used quantitatively"

Response: Corrected.

13. The statement "In the present study, . . .. . .. . .. . .. . .. . .." should start a new paragraph.

Response: Corrected.

14. Line 119: re-cast the statement beginning from "near the port". The near . . .. . .. . . near in the last part of the statement makes it ambiguous.

Response: Corrected.

15. Line 120: Is Kindo Peninsula in Arkhangelsk or Arkhangelsk is a sampling site on its own?

Response: Here, we clearly point to three different areas. We do not understand why the reviewer thinks that Kindo P. is in Arkhangelsk.

16. Line 121: should read ". . .. . .. . .. . .. . ... Pollution levels in these areas have been partly attributed to urban and gas flaring sources."

Response: Edited to better much reviewer's suggestions.

17. Line 125: should read ".................... to minimise the direct influence from ......."

Response: Corrected.

18. Line 126: should read ". . .. . .. . .. . .. . ... information about sample collection

such . . ..."

Response: Corrected.

19. Line 127: should read "................ and the depth at which snow was sampled ..."

Response: Corrected.

20. Line 129: should read "Sampling was perform with a metal-free technique using pre-cleaned ............................'

Response: Corrected.

21. Line 130: should read ".......... polyethylene bags which had been ................"

Response: Corrected.

22. Line 131: should read "............ 1M HCl and rinsed with abundant deionised ultra-pure water in the ........................"

Response: Corrected.

23. Line 133: should read " . . .. . .. . .. . ... filtered through 47 mm quartz fibre filters. The filters were dried at 60-70 oC ................................"

Response: Corrected but kept specifications of the filters in the manuscript.

24. Line 138-139: should read "Ele- mental carbon content of the filters were measured ................. (TOA) using the sunset laboratory . . .. . .. . .. . ..."

Response: Corrected.

25. Line 142: should read "Performance of the OC/EC instrument is regularly ........."

Response: Corrected.

26. Line 143-144: Recast leaving out the slash af- ter (EMEP).

Response: Corrected.

27. Line 148-149: should read "The carbonate content of filtrate on the filters was measured by TOA after thermal-oxidative . . .. .. . .. . .. . .. . ..."

Response: See major comments number 12.

28. Line 150: should read "A punch of 1.5 cm2 ......................."

Response: Corrected.

29. Line 152-154: Re-cast this sentence. Do you mean section 2.2 or chapter 2.2?

Response: Corrected. "Section" fits much better here.

30. Line 156 and 157: 'evolves' should be 'evolved'

Response: Corrected.

31. Line 158: should read "Applying this correction, EC values were ............"

Response: Corrected.

32. Line 160: Give the full meaning of LPDM at first use.

Response: The full meaning of LPDM is given at the last paragraph of Introduction.

33. Lines 163-165: Re-cast to read "The ECMWF data has 137 vertical data and a horizontal resolution of 1 x 1 for 2014 and 2015 simulation, and 0.5 x 0.5 for 2016"

Response: Corrected.

34. Line 188: mass per unit area you mean

Response: Corrected.

35. Could you re-cast this sentence?

Response: Perhaps the reviewer has forgotten adding the line where the sentence to be re-casted is.

36. Line 198-200: What are the rationale/references for these assumptions? Any similar assumption in literature?

Response: Of course. I re-write the response to a relevant comment from reviewer 1: BC particles in fresh exhaust are typically found in the 100 nm range or smaller and, in the urban environment, grow relatively quickly to sizes of about 200 nm (e.g., Ning et al., 2013). We agree that this occurs mainly via internal mixing with other types of aerosols. In remote areas, BC is mostly part of the internal aerosol mixture, with typical sizes of around 200 nm (see Freud et al., 2017, for Arctic size distributions). The wet diameters (which determine the physical behavior of the particles such as settling) will be larger than that. FLEXPART uses a single size distribution for BC aerosols and it does not account for particle growth. Therefore, a size distribution must be chosen that is representative for a broad range of conditions. Our size distribution is not representative for the external mixture of fresh BC particles (which are much smaller) but rather for the internal mixture of aerosols encountered in the Arctic and during most of the time BC resides in the atmosphere. It would not be appropriate to simulate the behavior of BC in fresh exhaust. Thus, while we totally agree with the reviewer about the mixing state of the BC particles, we think our settings are representative of this.

Freud, E., Krejci, R., Tunved, P., Leaitch, R., Nguyen, Q. T., Massling, A., Skov, H., and Barrie, L.: Pan-Arctic aerosol number size distributions: seasonality and transport patterns, Atmos. Chem. Phys., 17, 8101-8128, https://doi.org/10.5194/acp-17-8101-2017, 2017. Ning, Z., Chan, K.L., Wong, K.C., Westerdahl, D., Mocnik, G., Zhou, J. H., Cheung, C.S.: Black carbon mass size distributions of diesel exhaust and urban aerosols measured using differential mobility analyzer in tandem with Aethalometer, Atmos. Environ., 80, 31-40, 2013

37. Line 214: this should be Figure 1(c).

Response: Corrected.

38. Line 216: Like I stated in the general comment, you could report the 25th and 75th

percentile or 10th and 90th percentile.

Response: I think that the number of samples that we measured in not sufficient for that. We have maintained this presentation of concentrations (see major comment).

39. Line 221: should read "........the snow samples for 2014, EC con- centrations ....."

Response: Corrected.

40. Line 228: should read "................(on the White sea coast) showed high . . .. . ..."

Response: Corrected.

41. Line 232-240: Re-cast the five sentences in these lines.

Response: We have try to edit these sentences, but we do not know towards which direction as the comment is not very specific..

42. Line 239: should read " . . .. . .. . .. . .. . .. . .Tomsk and Yamal, EC concentration was highly .................................."

Response: Corrected.

43. Line 244: Should read "................measured EC concentrations in the snow sam- ples ........................"

Response: Corrected.

44. Line 246: A scatter plot of what? Figure 1 should be Figure 1(b).

Response: Corrected.

45. Line 247: should read ". . .. . .. . .. . .. . . agreement and good correlation . . .. . .. . .. . ...

Response: Edited.

46. Line 258: The sentence "The MFB of the . . .. . .. . .. . .. . .. . ... was -42%" is

somehow isolated. What inference can be drawn from the fact that MFB is -42%.

Response: Corrected.

47. Line 264: should read "For 2016, FB values ........................................... show another set of underestimation.

Response: Corrected.

48. Line 266: 12 out of 19 what? Samples?

Response: Corrected. Yes, we meant samples.

49. Line 266: Should read "19 samples. For the remaining 7 samples, the model ........"

Response: Corrected.

50. Line 267 should read "..................... The root mean square error . . .. . ."

Response: Corrected.

51. Line 268-269: Please, re-frame this sentence. The sentence, as it stands, is ambiguous. I guess it should read "The RMSE is fre- quently used to measure . . .. . .."

Response: Corrected.

52. Line 273-275: the sentence is muddled up. What exactly do you want the reader to infer from the two short sentences?

Response: Corrected.

53. Line 276-277: should read ". . ... reported that the maximum BC concentration measured ....................."

Response: Corrected.

54. Line 283: should read "...............Stockholm with a popula- tion of about 2 million.
Response: Corrected.

55. Line 287: What do you mean by one order of magnitude?

Response: Corrected.

56. Line 290: should read ". . .. . .. . ... Macdonald et al., (2017) reported BC concentrations ranging from ...................... For the samples collected near ............"

Response: Corrected.

57. Line 295: should read "......ECLIPSE emissions dataset .................." The word 'account' does not work here. Please, choice a different word.

Response: Corrected.

58. Line 297- 298: should read "........................... gas flaring (FLR) while biomass burning . . .. . .. . .. . .. . ..."

Response: Corrected.

59. Line 310-311: the list of cities in the bracket is just too long. In- clude only the important cities and move the bracket to immediate after 'major Russian cities on line 309.

Response: Corrected.

60. Line 313-316: re-cast this sentence to reflect what you want the reader to understand from the sentence,

Response: Corrected.

61. Line 319: should read "(6%) (see Figure 2)......................"

Response: Corrected.

62. Line 320-321: Are these two sources new? Where they not there in 2014?

Response: No, they are not new at all. It is simply the fact that samples collected from different regions are usually influenced by different sources.

63. Line 325-326: should read "........ Peninsula whereas FLR emissions ......... were very low due to the long distance of flaring emission sources from the sampling point."

Response: Corrected.

64. Line 327-328: should read ". . .. also affected BC concentration in snow in northwestern . . .. .. . .. . ..."

Response: Corrected.

65. Line 329: should read " releases in Russia, the miscalculation ............ and their impact in .............."

Response: Corrected.

66. Line 331-332: should read ". . .. . .. . .. . .BB emissions, originating mostly from eastern Europe, contributed about . . .. . .. . ..."

Response: Corrected.

67. Line 336: should read ". . ... Yamal, DOM, FLR TRA contributed, on the average, 31%, 29% and 27%, respectively (see Figure 2(c))."

Response: Corrected.

68. Line 341: it should be Figure 5(b) if you effect the comment on labelling of individual figures in the plot as suggested in the general comment section

Response: Corrected.

69. Line 353-359: Re-phrase the sentences on these lines stating what exactly you did will the data from Doherty and Macdonald as well as the reasons for the cross validation.

Response: Corrected.

70. Line 374-375: should read" . . .. . .. . .. . . Similar to our finding for the new Russian mea- surements, the model output, with a MFB of -51%, tends to underestimate deposition."

Response: Corrected.

71. Line 383-384: Are you referring to Doherty data here? If so, state that explicitly.

Response: It is now stated explicitly in the beginning of the paragraph (line 507).

72. Line 388: Expunge "Moreover" The sentence should read" Our model output was . . .with measured BC concentrations in . . .. . ."

Response: Corrected.

73. Line 392: 'research' should be 're- searcher'

Response: We believe not! It is a small population of research and military personnel.

74. Be explicit. Did you do a model run for the period for which Macdonald et al carried out measurements?

Response: Corrected. Please see line 545.

75. Line 401: underestimated what?

Response: Corrected.

76. Line 402: should read "Further analysis was carried out to adequately understand . . .."

Response: Corrected.

77. Line 404-408: Re-cast this complex sentence into 2 – 3 simple ones.

Response: Corrected.

78. Line 413: should read ".............Two hotspots were ......."

Response: Corrected.

79. Line 414: should read ".... And an- other, of smaller intensity, in southeastern Asia."

Response: Corrected.

80. Line 415-417: The two simple sentences here are disjointed.

Response: Corrected.

81. Line 419: should read". . .. . ... America in ECLIPSE. The Alert samples, for which the model strongly underestimated BC, the major sources . . .. . .. . .. . ...".

Response: Corrected.

82. Line 421: Why is 7 ng g-1 not in percentage?

Response: Corrected.

83. Line 422: should read". . .. . .. . .. Alert air pollutant concentrations . . .. . .. . .."

Response: Corrected.

84. Line 429: should read "It has been shown that average measured . . .. . .. . ..."

Response: Corrected.

85. Line 432: delete 'already.'

Response: Corrected.

86. Line 437: should read". . .. . .. . .. . . locations of fires that have been active in the last two months before the sample collection. The fire data were adopted from MODIS . . .."

Response: "Active fires" is a very common product of MODIS and this is the reason that we want to keep this expression. The rest has been corrected according to the

reviewer's suggestion.

87. Line 439: gas flaring facilities or gas flaring data?

Response: It is "gas flaring facilities" what we plot in Figure 7.

88. Line 443: How do you mean? Around gas flaring facilities?

Response: Khanty-Mansijsk region is known among scientists that study BC transport as one of the most important regions of gas-flaring emissions in the world. Yes, there are many facilities of this type in the area, and they can be easily seen from space (see VIIRS data in Fig. 1): Circle shows Nenets-Komi and rectangle Khanty/Mansijsk regions.

89. Line 445: should read". . .. . ... According to a related study by Huang and Fu (2016), . . .. . .. . .."

Response: Corrected.

90. Line 450: Which model are you refer- ring to here?

Response: Corrected.

91. Line 451-452: These cities/regions are not explicitly labelled in the plots. So that the reader can follow through with the discussions, it is better to include lon/lat of these cities/regions in a bracket. Could you do this for other locations in sim- ilar discussion throughout the manuscript at their first mention?

Response: These regions mentioned are well known among scientists that study BC. E are talking about the most important global sources of BC located inside the Polar Dome, which directly affect the Arctic. We believe that adding so much information in the figures will put a lot of pressure on how to make everything visible to the readers (cities, regions of interest, colors representing data, etc. . .).

92. Line 452: should be Figure 7(b)

Response: Corrected.

93. Line 458-459: should read". . .. . .. northwestern Russia, a region which includes Murmansk. Pollution level in Murmansk could be high due to . . .. . .. . ..."

Response: Corrected.

94. Line 462: You have referred to figure 7 severally but these cities are not explicitly shown in Figure 7.

Response: See response in comment 91. These regions are easily seen by the hotspots that are visible in Figure 7.

95. Line 467: should read". . .. . .. . ... polluting sources identified in ECLIPSE dataset."

Response: Corrected.

96. Line 471: should read "to have originated mainly . . .. . .. . .. . .."

Response: Corrected.

97. Line 472-473: Re-cast this sentence. Insignificant? Negligible?

Response: Corrected.

98. Line 474: delete (Figure S5). It makes no contribution to the sentence.

Response: Moved below.

99. Line 475: should read". . ... European Russia (Figure S5)"

Response: Corrected.

100. Line 476: should read". . .. . ..of the total contribution, which reflect the proximity of the sampling site to the main flaring facilities in Russia."

Response: Corrected.

101. Line 477-479: could you re-phrase this sentence?

Response: Corrected.

102. Line 480: delete Figure (S6). It adds nothing to the understanding of this sentence,

Response: Moved below.

103. Line 482: Delete 'Again' sub-categories should be 'categories.'

Response: Corrected.

104. Line 483-484: What could be responsible for the insignificant contribution of FLR at this sampling site? Is the site upwind of the flaring facilities? It would be interesting if you could put forward an argument for this insignificant contribution despite the closeness to the flaring facili- ties.

Response: Corrected.

105. Line 499: : should read"........... Russia in 2014, 2015 and 2016 ....... EC concentration"

Response: Corrected.

106. Line 501-502: should read". . ... to measured BC concentration in snow . . .."

Response: Corrected.

107. Line 507: : should read". . .. . .. Russian emission as well as . . .. . ..."

Response: Corrected.

108. Line 515: should read ". . .. . .. . .. . .. emissions originating from highly. . .. . .."

Response: Corrected.

109. Line 525-526: should read". . ... Considering the fact that similar . . .. . .. . .. .

.observed in samples collected in the area during other years, it is likely . . . of BC in this region show . . .. . .. . .."

Response: Corrected.

110. Line 528: should read". . .. previously reported average measurements of BC concentrations in snow in Western . . .. . ..."

Response: Corrected.

111. Line 529: delete 'on average'.

Response: Corrected.

112. Line 584: Delete one of the 'doi'.

Response: Corrected.

113. Line 623-626: This reference is cited as 2016 in the manuscript (see line 207).

Response: Corrected.
* * *
[Figure]

[Figure]

**Fig. 1.**

---

## Author Response (AR2)

Co-Editor Decision: Publish subject to minor revisions (review by editor) (14 Dec 2017) by Rob MacKenzie

Comments to the Author:

Dear Dr Evangeliou and colleagues,
Thank you for your patience; we have struggled to find referees with time to provide a review, but I am glad that we have in the end got two informative reviews. I am pleased to accept your paper for ACP subject to the revisions outlined below, which derive from reviewers comments that I don't think you have quite fully satisfied yet,
Sincerely,
Rob MacKenzie

**Response**: We appreciate Editor's efforts to get reviews as fast as possible. We have tried to satisfy the comments that Editor considers so important in order our article to be published in ACP.

Abstract and elsewhere: to avoid the potential for misunderstandings, I suggest reporting the modulus of the FB accompanied by the word 'underestimate'. You should report the RMSE results as well as FB in the abstract.

**Response**: Corrected! See manuscript with track changes in abstract and elsewhere.

Lines 50-52: I don't think you can use the terms "quite accurately" and "small discrepancies" when you have already reported |FB|>100% a few lines above. Please find a more specific summary of the model evaluation.

**Response**: Corrected! See line 52.

Lines 52-54. I don't see how this final sentence follows from what went before. I suggest you state that your independent datasets were 50% lower than your own measurements at line 50.

**Response**: We agree that this sentence does not follow what is stated above and we have removed it. Paragraph now ends with line 53.

Line 80 ppb here is presumably mass/mass (ng/g). Better to be explicit, I think, so as not to hold up the reader.

**Response**: Corrected! Please see line 88.

In the discussion on lines 120-124, and in Figure 1, please refer the reader to Table S1.

**Response**: Corrected! Please see paragraph in lines 129-141.

For Figure 1c, please refer to Silverstein et al. (2008) and try to find a better colour map.

**Response**: Corrected! Though, we do not understand how a publication entitled "Automatic Perceptual Color Map Generation for Realistic Volume Visualization" is related with a plot that uses default colorbars from the matplotlib library of the open access Python language.
We have included the reference in the manuscript (line 879) and we have tried to use another colorbar for Fig. 1(c) (see line 873).

Line 180. Do you really mean that trajectories were released at 20km altitude? This is well inside the polar stratosphere and so very unlikely to contribute to wet deposition at the trajectory endpoints.

**Response**: The right statement is that the computational particles were released between 0 and 20,000 meters.
We know that 20 km is a stratospheric altitude. However, from tests that we have made, we have found that this is the optimal altitude in order to account for all wet scavenging processes that occur in the atmosphere. I admit that if we release the same computational particles within 0-8,000 meters or less, we may end up with the same results. It does not cost anything to us (except for some additional computational time) to account for higher altitudes though. Please see all the tests made in https://www.geosci-model-dev.net/10/4605/2017/.

Line 217 or thereabouts. Please insert a short subsection prefacing your approach to the statistics you will use to report measurements and the statistics you will use to compare model and measurements. You can move the material around line 260 to here to consolidate. This way you will set out your approach ahead of reporting the results. Readers may not agree completely with your approach, but at least they will know what it is and get used to it. In the current version, queries about the statistical approach get in the way of considering the scientific implications of the actual results. I would encourage you to draw the reader's attention to figures such as S1 and S3 in this discussion, because these show clearly that summary statistics of central tendency and RMSE need to be interpreted with care. Figure S3 particularly shows the well-known property of the correlation coefficient that it can report a high value (goodness of fit) for an observation vs model scattergram with gradient very different from unity.

**Response**: Corrected! We have added a paragraph after section 3 about the statistical tools that we have used. We have highlighted Figures S1 – S3 as the figures that summarize our findings (lines 232-245).

Line 223 and elsewhere. Please don't mix resistant and robust statistical measures (percentiles) with non-resistant and non-robust measures (range, mean, and standard deviation). Report mean together with standard deviation, and median together with quartiles. It is perfectly possible to estimate percentiles for a sample of 10 or 11 – see a standard statistics textbook, or http://www.itl.nist.gov/div898/handbook/prc/section2/prc262.htm for a quick hint.

**Response**: We agree with this comment and we have corrected the manuscript wherever needed. Please see track changes in the manuscript.

Lines 268-270. The discussion of FB gets very convoluted here, with positive and negative numbers in close proximity. I realise it's awkward, but please look at simplifying, perhaps by talking about the modulus of FB.

**Response**: Corrected! We have tried to do this wherever it was possible and we have also reformulated many sentences in this direction. Please see track changes in the manuscript.

Lines 296-300. Please report medians and percentiles from the literature, rather than just upper limits.

**Response**: In this section, we report concentrations from the literature as the reviewer has pointed out. Although we have all the data presented in Doherty et al. or MacDonald et al. papers and we could perform any kind of statistics, we have not all the observations from Aamaas et al. (2011), Ruppel et al. (2014), McConnel et al. (2007), Forsstrom et al. (2013) or Svensson et al. (2013). Since we report values as ranges (Min–Max concentrations) for all the aforementioned studies, we think that it would be ackward to report median +- quartiles for Deherty et al. (2010) and MacDonald et al. (2017). For better consistency in the current paragraph, we would like to keep ranges for all the measurements collected from the relevant literature. We hope that the Editor will understand and agree.

Line 392. Remove double full stop.

**Response**: Corrected (line 473).

Line 402. Representing the Doherty et al. (2010) data as a mean with standard deviation leads you to reporting a standard deviation bigger than the mean. For data that is positive definite this simply illustrates the difficulties of working with assumed Gaussian statistics on highly skew data. I suggest you replace with median and interquartile range.

**Response**: This problem of reporting higher standard deviations that the mean can also occur when presenting median and interquartile ranges (see, for instance, corrected values for our campaign in 2014). We have substituted mean+-sd with median+-interquartile ranges! See lines 485-487, and 493-494.

Lines 414-415. I would suggest replacing "no safe conclusions" with a statement that it is not clear whether the discrepancy arises as a measurement artefact (even though every effort has been taken in both papers to follow a robust protocol) or from real spatio-temporal variation.

**Response**: Corrected. See lines 498-500.

Reference
Silverstein, J., Parsad, N., and Tsirline, V. (2008). Automatic perceptual color map generation for realistic volume visualization. Journal of Biomedical Informatics, 41(6):927.

[revised manuscript text omitted]

Nikolaos Evangeliou 18/12/2017 16:50

Nikolaos Evangeliou 18/12/2017 16:50

Nikolaos Evangeliou 18/12/2017 16:49

Nikolaos Evangeliou 18/12/2017 16:49

Nikolaos Evangeliou 18/12/2017 16:58

Nikolaos Evangeliou 18/12/2017 17:00

Nikolaos Evangeliou 18/12/2017 16:58

Nikolaos Evangeliou 18/12/2017 16:58

Nikolaos Evangeliou 18/12/2017 16:58

Nikolaos Evangeliou 18/12/2017 17:00

Nikolaos Evangeliou 18/12/2017 17:00

Nikolaos Evangeliou 18/12/2017 17:01

Nikolaos Evangeliou 18/12/2017 17:01

Nikolaos Evangeliou 18/12/2017 17:04

Nikolaos Evangeliou 18/12/2017 17:04

Nikolaos Evangeliou 18/12/2017 17:06

Nikolaos Evangeliou 18/12/2017 17:15

[revised manuscript text omitted]